# One Model to Drift Them All: Physics-Informed Conditional Diffusion Model for Driving at the Limits

**Franck Djeumou**[1,2], **Thomas Lew**[1], **Nan Ding**[1], **Michael Thompson**[1],
**Makoto Suminaka**[1], **Marcus Greiff**[1], and **John Subosits**[1]
[1]`Toyota Research Institute`, [2]`Rensselaer Polytechnic Institute`

**Abstract:** Enabling autonomous vehicles to reliably operate at the limits of handling— where tire forces are saturated — would improve their safety, particularly in scenarios like emergency obstacle avoidance or adverse weather conditions. However, unlocking this capability is challenging due to the task's dynamic nature and the high sensitivity to uncertain properties of the road, vehicle, and their dynamic interactions. Motivated by these challenges, we propose a framework to learn a conditional diffusion model for high-performance vehicle control using an unlabelled dataset containing trajectories from distinct vehicles in different environments. We design the diffusion model to capture the complex dataset's trajectory distribution through a multimodal distribution of parameters of a physics-informed data-driven dynamics model. By conditioning the generation process on online measurements, we integrate the diffusion model into a real-time model predictive control framework for driving at the limits, and show that it can adapt on the fly to a given vehicle and environment. Extensive experiments on a Toyota Supra and a Lexus LC 500 show that a *single* diffusion model enables reliable autonomous drifting on both vehicles when operating with different tires in varying road conditions. The model matches the performance of task-specific expert models while outperforming them in generalization to unseen conditions, paving the way towards a general, reliable method for autonomous driving at the limits of handling.

**Keywords:** Diffusion Models, Learning for Control, Autonomous Drifting.

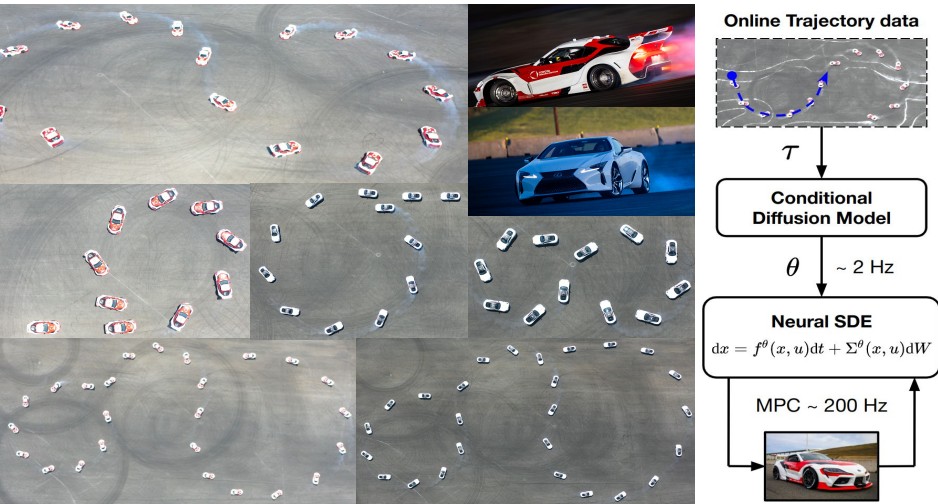

Figure 1: Left: Examples of the conditional diffusion model performing drifting trajectories on two vehicles. Right: Overview of the controller architecture and online model parameter generation process. The videos of the experiments can be found at `https://tinyurl.com/diff-drift`.

## 1 Introduction

Existing autonomous vehicles are constrained to operate at a fraction of their full handling potential. Designing algorithms to reliably control vehicles beyond these engineered limits would unlock faster

8th Conference on Robot Learning (CoRL 2024), Munich, Germany.

and more reliable responses to diverse safety-critical situations [1, 2] such as driving on ice and avoiding sudden obstacles, scenarios where the vehicle may use all of the available tire-road friction, causing it to slide across the road [3, 4, 5]. However, driving at the limits of handling is challenging due to the task's dynamic nature, the high sensitivity to model mismatch, and the uncertain properties of the road, the vehicle, and their dynamic interactions. In addition, the high cost of collecting a dataset for driving at the limits, the complex vehicle dynamics, and the safety considerations complicate the use of imitation learning and reinforcement learning strategies. These challenges motivate the development of a model capable of exploiting physics knowledge and capturing complex forms of uncertainties while being amenable to real-time autonomous vehicle control.

Diffusion models [6, 7, 8, 9] have shown to be highly capable of representing complex, high-dimensional, and multimodal distributions from data. However, their direct use for driving at the limits is not straightforward. The limitations of classical diffusion models include the question of how to leverage prior physics knowledge to improve data efficiency and interpretability, and the considerable model inference time, which can be a bottleneck for high-bandwidth control.

**Contribution.** We propose a conditional diffusion vehicle model for control-oriented modeling of driving at the limits of handling under uncertainties. By predicting the parameters of a physics-informed neural stochastic differential equation dynamics model, the model has four key properties.

- By encoding prior physics knowledge as an inductive bias, the proposed diffusion-based vehicle model is interpretable and generalizes to new environments from small amounts of data.

- The model can capture complex multimodal distributions over the vehicle model's parameters.

- The model can adapt on the fly to various test-time vehicles and road conditions by conditioning on measurements of the vehicle's interaction with the world.

- By predicting the parameters of a physics-based model, as opposed to directly predicting state trajectories, the model inference and control loops are decoupled. This hierarchical approach unlocks diffusion sampling at low rates and high-frequency predictive control.

We integrate the diffusion model in a real-time nonlinear model predictive control framework for autonomous driving at the friction limits, and extensively validate it on a Toyota Supra and a Lexus LC500. Our results showcase that a single diffusion model can reliably control both vehicles on challenging drifting tasks involving different road conditions and vehicle properties, see Figure 1.

## 2 Related work

Several works have explored autonomous driving at the limits of handling, both in the context of racing [10, 11, 12, 13] and drifting [14, 5, 4, 15]. These methods identify the parameters of a physics-based vehicle dynamics model [16, 17, 4, 18, 19] or train a neural network model [15, 20], and subsequently use it for model-based optimal control. Nonlinear model predictive control (MPC) is the go-to control strategy in such settings and has demonstrated high-performance tracking ability in challenging racing and drifting tasks [12, 11, 4, 15]. However, the performance of MPC is limited by the fidelity of the vehicle model, designed to capture a single vehicle with given tires operating in specific road conditions. In contrast, using an unlabeled trajectory dataset, we train a single generative vehicle model with online adaptation capabilities that enable autonomous driving at the limits of handling on different vehicles in varying road conditions.

Diffusion models have emerged as a powerful tool for generating complex and multimodal distributions in continuous domains such as images [21, 22], 3D contents [23, 24], planning and control [25, 26, 27, 28, 29, 30, 31], time series [32, 33], and physics processes [34, 35, 36]. However, all these works learn to represent, in a black-box manner, distributions for which samples are directly available in the training data. In contrast, we train a diffusion model to generate samples from a latent space of vehicle dynamics parameters that are not in the training data. Our approach relates to research in latent diffusion models [23, 37]. But instead of learning encoder and decoder networks to map between the data and latent spaces, we impose a structure on the latent space through physics-informed neural stochastic differential equations.

Neural ordinary differential equations [38, 39, 40, 41, 42], Koopman operators [43, 44, 45, 46, 47, 48], classical system identification [49, 50, 51, 52, 53], Gaussian processes [54, 55, 56, 57, 58], and neural stochastic differential equations (SDEs) [59, 60, 61, 62, 63] have been widely studied for modeling uncertain dynamical systems from data. These models are either deterministic or can capture only a

single mode of the training dataset's distribution, along with the uncertainty around the mode. On the other hand, Bayesian inference on the parameters of some of these models can, in theory, capture the multimodal distribution of the model parameters. However, classical Monte Carlo-based methods [64, 65, 66, 67] do not scale well with large models and datasets, while variational inference-based methods [68, 69, 70] are limited by the choice of variational family used to approximate multimodal posteriors. To address these limitations, recent works [71, 72, 73, 36] have highlighted the scalability and expressivity of diffusion models when approximating the posterior distribution in Bayesian inference. We leverage such expressivity to capture a multimodal distribution over the parameters of a vehicle model expressed as a neural SDE: A model shown in [63] to improve long-term prediction accuracy and uncertainty estimate compared to deep Gaussian-based models [74, 75].

Our modeling approach is similar to meta-learning methods [76, 77, 78] since the diffusion model learns offline to predict the parameters of a neural SDE model while adapting online to different vehicles and environment conditions. However, in contrast to existing meta-learning approaches where the dataset has task-specific labels, we train our model from an unlabeled dataset of vehicle trajectories. In addition, while conditioning the offline-trained diffusion model enables online adaptation, we emphasize that no gradient updates or any sort of regression on the model parameters are performed online for adaptation, as is typically done in meta-learning or online learning [79, 80, 81, 82, 83].

## 3 Method

We assume access to a dataset $\mathcal{T} = \{\tau_1, \ldots, \tau_{|\mathcal{T}|}\}$ of vehicle trajectories $\tau = \{(x_{t_0}, u_{t_0}), \ldots, (x_{t_{|\tau|}}, u_{t_{|\tau|}})\}$, where each $(x_t, u_t)$ denotes a state-control pair. We denote by $x_{t:T_f} = [x_t, \ldots, x_{t+T_f}]$ and $u_{t:T_f} = [u_t, \ldots, u_{t+T_f}]$ the future state and control sequence from time $t$ to time $t + T_f$, and by $x_{T_p:t}$ and $u_{T_p:t}$ the past sequence from $t - T_p$ to $t$. We also use $\tau_{t:T_f}$, $\tau_{T_p:t}$, and $\tau_{T_p:t:T_f}$ to denote the future, past, and full sequence of state-control pairs at time $t$,

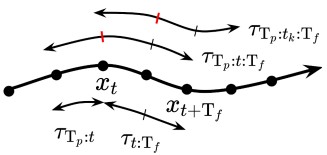

Figure 2: Trajectories in $\mathcal{T}$.

respectively. We consider that the dataset's trajectories may be collected from vehicles with different physical and tire specifications, and operating on various road conditions while performing different tasks. Thus, the distribution over its trajectories is complex and challenging to model. The dataset is also characterized as unlabelled since no information (per trajectories) about the vehicles and environments is provided for model identification other than the state-control pairs.

We present our approach to learning physics-constrained generative models for autonomous driving at the limits. The main idea is to integrate the structure and the existing physics knowledge of driving at the limits into the design of a neural SDE model parametrized by $\theta$. Then, we train a state-control history conditioned diffusion model, parameterized by $\psi$, to output a distribution $\mathrm{p}_\psi(\theta|x_{T_p:t}, u_{T_p:t})$ over the neural SDE parameters $\theta$, which is then used to generate the future state trajectories or models to use by an MPC controller. However, existing approaches to training diffusion models cannot be directly applied here as they would require unavailable access to a dataset of neural SDE parameters. The proposed approach (Section 3.1) generates such a dataset and consistently improves its parameters with respect to the trajectories in $\mathcal{T}$ while simultaneously training the diffusion model.

**Neural SDEs for modeling.** Neural SDEs [84, 85, 59, 60, 63] offer a principled approach for modeling uncertain dynamical systems due to their ability to encode prior physics knowledge from first principles, their calibrated uncertainties, and their expressiveness from using neural networks:

$$\mathrm{d}x = f^\theta(x, u)\,\mathrm{d}t + \Sigma^\theta(x, u)\mathrm{d}W, \tag{1}$$

where $x \in \mathcal{X} \subseteq \mathbb{R}^{n_x}$ is the state, $u \in \mathcal{U} \subseteq \mathbb{R}^{n_u}$ is the control input, $t$ is the time, $W$ is the $n_x$-th dimensional Wiener process, and $f^\theta : \mathcal{X} \times \mathcal{U} \to \mathbb{R}^{n_x}$ and $\Sigma^\theta : \mathcal{X} \times \mathcal{U} \to \mathbb{R}_+^{n_x \times n_x}$ are the parametrized drift and diffusion terms, respectively, and the equation is interpreted as an SDE in the Itô sense. Under smoothness assumptions [86, 87] on $f^\theta, \Sigma^\theta$, we can efficiently sample a distribution of predicted trajectories via numerical integration of the neural SDE [88, 89]. Assuming approximate Gaussian transitions between discrete times of a numerical SDE integrator, the negative log-likelihood (NLL) loss of a state sequence $x_{t:T_f}$ given the controls $u_{t:T_f}$ can be estimated by

$$\mathcal{J}_{\mathrm{nll}}(\theta, \tau_{t:T_f}) := \mathbb{E}_{\tilde{x}_{t:T_f}^\theta} \left[ \sum_{s=t}^{t+T_f} \|x_s - \tilde{x}_s^\theta\|_{(\Sigma_s^\theta)^{-1}}^2 + \log\left(\det\left(\Sigma_s^\theta\right)\right) \right], \tag{2}$$

with $\|z\|_A^2 := z^\top A z$ for $z \in \mathbb{R}^{n_x}$, $A \in \mathbb{R}^{n_x \times n_x}$, and where $\Sigma_s^\theta := \Sigma^\theta(\tilde{x}_s^\theta, u_s)$ and $\tilde{x}_{t:T_f}^\theta$ is a sample sequence obtained from the SDE integration for a fixed $\theta$ and the initial conditions $x_t$ and $u_{t:T_f}$.

**Algorithm 1** Training of the Diffusion Model $\psi$

1: Initialize $\mathcal{D} = \emptyset$.
2: Compute $\theta^{\mathrm{loc}}$ by solving (3) using SGD.
3: **while** not converged **do**
4:     Sample $\tau_{\mathrm{T_p}:t:\mathrm{T_f}}$ from $\mathcal{T}$.
5:     Sample $\{\tau_{\mathrm{T_p}:t_k:\mathrm{T_f}}\}_k$ around $\tau_{\mathrm{T_p}:t:\mathrm{T_f}}$.
6:     **if** Refine **then**
7:         Update $\theta^{\mathrm{loc}}$ with $\mathrm{p}_\psi(\cdot|h(\tau_{\mathrm{T_p}:t}))$.
8:     **end if**
9:     Optimize for $\theta_t$ using (4) and $\theta^{\mathrm{loc}}$.
10:     Append $(\tau_{\mathrm{T_p}:t}, \theta_t)$ to $\mathcal{D}$.
11:     Update $\psi$ using (5) on batches from $\mathcal{D}$.
12: **end while**

**Algorithm 2** Online MPC Model Sampling

1: Initialize history dataset $\mathcal{T}_{\mathrm{hist}} = \emptyset$.
2: Initialize best parameters $\theta^1, \ldots, \theta^{n_{\mathrm{best}}}$.
3: **while** not terminated **do**
4:     Append latest $(x_t, u_t)$ to $\mathcal{T}_{\mathrm{hist}}$.
5:     $\mathcal{T}_{\mathrm{gen}} \leftarrow \{\tau_{\mathrm{T_p}:t_k}\}_k$, $\mathcal{T}_{\mathrm{val}} \leftarrow \{\tau_{t_j:\mathrm{T_f}}\}_j$ are subsets of $\mathcal{T}_{\mathrm{hist}}$
6:     $\{\theta_{t_k}\}_k \sim \mathrm{p}_\psi(\cdot|\{\tau_{\mathrm{T_p}:t_k}\}_k)$ using (7).
7:     Evaluate scores $\mathcal{J}_{\mathrm{traj}}(\theta, \mathcal{T}_{\mathrm{val}})$ for $\theta \in \{\theta_{t_k}\}_k \cup \{\theta^p\}_{p=1}^{n_{\mathrm{best}}}$.
8:     Update $\Theta := \{\theta^p\}_{p=1}^{n_{\mathrm{best}}}$ with best scores.
9:     Send $\theta_{\mathrm{best}} = \mathrm{argmin}_{\theta \in \Theta} \mathcal{J}_{\mathrm{traj}}(\theta, \mathcal{T}_{\mathrm{val}})$ to MPC.
10: **end while**

## 3.1 Conditioned diffusion model in parameter space

We provide the main steps of the training process below and summarize them in Algorithm 1.

**Initial estimate of the neural SDE parameters.** First, we compute the maximum a posteriori estimate of the neural SDE parameters $\theta$ over the entire dataset $\mathcal{T}$. The training problem is given by

$$\theta^{\mathrm{loc}} \leftarrow \mathrm{argmin}_\theta \mathcal{J}_{\mathrm{traj}}(\theta, \mathcal{T}), \text{ with } \mathcal{J}_{\mathrm{traj}}(\theta, \mathcal{T}) \coloneqq \mathbb{E}_{(\tau_{t:T_f} \sim \mathcal{T})}\big[\mathcal{J}_{\mathrm{nll}}(\theta, \tau_{t:T_f})\big] + \lambda_{\mathrm{traj}}\mathcal{R}(\theta), \quad (3)$$

where $\mathcal{J}_{\mathrm{nll}}$ is the negative log-likelihood defined in (2), and $\tau \sim \mathcal{T}$ is a sampled trajectory from the dataset. The term $\lambda_{\mathrm{traj}}$ controls the regularization term $\mathcal{R}(\theta)$ that enforces available prior knowledge on the neural SDE parameters; see Appendix A.4 for details on the design choices in this section.

**Parameter dataset generation via local optimization.** Next, we use the estimate $\theta^{\mathrm{loc}}$ to iteratively generate a dataset $\mathcal{D} = \{(\tau_{\mathrm{T_p}:t}, \theta_t), \ldots\}$ of short state-control trajectories $\tau_{\mathrm{T_p}:t}$ and model parameters $\theta_t$ that are consistent with (a) the data sequence $\tau_{\mathrm{T_p}:t}$, (b) the immediate future sequence $\tau_{t:\mathrm{T_f}}$, and (c) additional sequences $\tau_{\mathrm{T_p}:t_k:\mathrm{T_f}}$ in a neighborhood of the initial data sequence $\tau_{\mathrm{T_p}:t:\mathrm{T_f}}$. The timesteps $t_k$ are sampled from a uniform distribution $\mathbb{U}$ of fixed width centered at $t$. The local optimal parameter $\theta_t$ that is added to the parameters dataset $\mathcal{D}$ is given by $\theta_t = \mathrm{argmin}_\theta \mathcal{J}_{\mathrm{loc}}(\tau_{\mathrm{T_p}:t}, \theta, \theta^{\mathrm{loc}})$ with

$$\mathcal{J}_{\mathrm{loc}}(\tau_{\mathrm{T_p}:t}, \theta, \theta^{\mathrm{loc}}) \coloneqq \mathcal{J}_{\mathrm{nll}}(\theta, \tau_{t:\mathrm{T_f}}) + \mathbb{E}_{t_k \sim \mathbb{U}_{(t-\mathrm{W}, t+\mathrm{W})}}\big[\mathcal{J}_{\mathrm{nll}}(\theta, \tau_{\mathrm{T_p}:t_k:\mathrm{T_f}})\big] + \lambda_{\mathrm{loc}}\|\theta - \theta^{\mathrm{loc}}\|^2, \quad (4)$$

where the term in $\lambda_{\mathrm{loc}}$ regularizes $\theta$ to be close to the estimate $\theta^{\mathrm{loc}}$. The additional sequences $\tau_{\mathrm{T_p}:t_k:\mathrm{T_f}}$ help refine the uncertainty estimates of the neural SDE model. Using short sequences $\tau_{\mathrm{T_p}:t_k:\mathrm{T_f}}$, with $t_k$ in a small time window $\mathbb{W} \in \mathbb{R}_+$ around $t$, helps ensure that they can be explained by a single parameter vector $\theta_t$, since properties of the system may otherwise vary over a trajectory $\tau_{\mathrm{T_p}:t:\mathrm{T_f}}$ if $\tau_{\mathrm{T_p}:t:\mathrm{T_f}}$ is too long. We optimize for $\theta_t$ using gradient descent on (4) starting from an initial guess sampled from a Gaussian distribution centered at $\theta^{\mathrm{loc}}$. We found that regularizing to the estimate $\theta^{\mathrm{loc}}$ in the objective (4) stabilizes the optimization process.

**Training the diffusion model.** Given the generated neural SDE parameters in $\mathcal{D}$, we update the parameters $\psi$ of our conditional denoising diffusion model with gradient descent as in [7], on a loss

$$\mathcal{J}_{\mathrm{DM}}(\psi) = \mathbb{E}_{(\theta_t, \tau_{\mathrm{T_p}:t}) \sim \mathcal{D}, \epsilon \sim \mathcal{N}(\mathbf{0}, \mathbf{I}), k \sim \mathbb{U}_{(1, K)}}\big[\|\epsilon_\psi(\sqrt{\gamma_k}\theta_t + \sqrt{1 - \gamma_k}\epsilon, k, h(\tau_{\mathrm{T_p}:t})) - \epsilon\|^2\big], \quad (5)$$

where $\gamma_k \coloneqq \prod_{i=1}^k (1 - \beta_i)$ with $\beta_i \in (0, 1)$ being a linear noise schedule to gradually distort the parameters dataset $\mathcal{D}$, $\mathcal{N}(\mathbf{0}, \mathbf{I})$ is the standard normal distribution, $K$ is the number of diffusion steps, and $\epsilon_\psi$ is the denoising neural network predicting the noise $\epsilon$ added during the noising process. Here, $h$ is a function that maps the history $\tau_{\mathrm{T_p}:t}$ to a feature space for conditioning the diffusion model:

$$h(\tau_{\mathrm{T_p}:t}) = [\Delta x_{\mathrm{T_p}:t}\Delta t_{\mathrm{T_p}:t}^{-1}, u_{\mathrm{T_p}:t}], \text{ with } \Delta x_{\mathrm{T_p}:t} = [x_{\mathrm{T_p}+1} - x_{\mathrm{T_p}}, \ldots, x_t - x_{t-1}], \quad (6)$$

and $\Delta t_{\mathrm{T_p}:t}^{-1} = [(t_{\mathrm{T_p}+1} - t_{\mathrm{T_p}})^{-1}, \ldots, (t_t - t_{t-1})^{-1}]$. We use $K = 1000$ noising steps as in [7].

**Iteratively refining the parameter dataset.** As the diffusion model training progresses, its predictions of the local parameters $\theta_t$ become more accurate while the initial estimate $\theta^{\mathrm{loc}}$ computed in (3)

may become a suboptimal initial guess for $\theta_t$. Thus, after some number of diffusion training steps, we use the generative model to refine the dataset $\mathcal{D}$ by sampling a neural SDE parameter that would serve as an initial estimate $\theta^{\mathrm{loc}}$ and a regularizer for the local optimization problem (4).

**Online diffusion model inference.** Algorithm 2 outlines a simple online strategy for sampling the neural SDE parameters conditioned on online measurements. The algorithm maintains the history of state-control pairs and uses it as a source for generating, validating, and scoring the neural SDE parameters. At each time step of the algorithm, a set $\mathcal{T}_{\mathrm{gen}}$ of state-action sequences is sampled from the history dataset $\mathcal{T}_{\mathrm{hist}}$ to generate a set of parameters $\theta_t$ conditioned on each $\tau_{\mathrm{T_p}:t} \in \mathcal{T}_{\mathrm{gen}}$. This is done in $K$ steps, by sampling $\bar{\theta}_K \sim \mathcal{N}(0, \mathbf{I})$ and refining $\bar{\theta}_{k-1} \sim \mathcal{N}(\mu_\psi(\bar{\theta}_k, k, h(\tau_{\mathrm{T_p}:t})), \sigma_k \mathbf{I})$ with

$$\sigma_k = \beta_k \frac{1 - \gamma_{k-1}}{1 - \gamma_k}, \text{ and } \mu_\psi(\bar{\theta}_k, k, z) = \frac{1}{\sqrt{1 - \beta_k}}(\bar{\theta}_k - \frac{\beta_k}{\sqrt{1 - \gamma_k}}\epsilon_\psi(\bar{\theta}_k, k, z)), \qquad (7)$$

before letting $\theta_t = \bar{\theta}_0$. Finally, a set $\mathcal{T}_{\mathrm{val}} \subseteq \mathcal{T}_{\mathrm{hist}}$ is sampled and used to validate the generated parameters and compute their scores, defined as the loss $\mathcal{J}_{\mathrm{traj}}(\theta, \mathcal{T}_{\mathrm{val}})$ in (3) plus a 2-norm regularization term that penalizes the distance to the $n_{\mathrm{best}} = 5$ best previously generated parameters $\{\theta^p\}_{p=1}^{n_{\mathrm{best}}}$. We found that this last term helps ensure consistent updates during the online inference process.

### 3.2 Application to autonomous driving at the limits of handling

**Physics-constrained neural SDE model.** We now introduce the uncertainty-aware and physics-constrained neural SDE model for driving at the limits of handling. We employ the commonly used single-track model [90, 16, 91, 18, 17] as a foundation to describe the nonlinear dynamics of the vehicle. The vehicle position is expressed in a curvilinear coordinate system relative to a reference trajectory [14, 19, 12]. Specifically, the position coordinate is described by the distance $s$ along the reference trajectory, the relative heading $\Delta\phi$ with respect to a reference heading $\phi_{\mathrm{ref}}$, and the lateral deviation $e$ from the path. The proposed neural SDE model is given by

$$\mathrm{d}x = \mathrm{M}^\theta(x, u)F^\theta(x, u)\mathrm{d}t + \Sigma^\theta(x, u)\mathrm{d}W, \qquad (8)$$

where prior knowledge comes from the matrix $\mathrm{M}^\theta(\cdot, \cdot)$ that depends on vehicle parameters such as the mass $m^\theta$, yaw moment of inertia $I_z^\theta$, rotational inertia of the drivetrain $I_w^\theta$, tire radius $R^\theta$, and distances from the center of gravity to the front and rear axles $a^\theta$ and $b^\theta$. The control input $u = [\delta, \tau^{\mathrm{e}}]$ is the steering angle and engine torque, respectively. The state $x = [r, V, \beta, \omega_r, e, \Delta\phi, s]$ includes the yaw rate $r$, velocity $V$, sideslip angle $\beta$, rear wheelspeed $\omega_r$, lateral error $e$, and angular deviation $\Delta\phi$. Lastly, $F^\theta = [F_{xf}^\theta, F_{yf}^\theta, F_{xr}^\theta, F_{yr}^\theta]$ represents the tire forces between the vehicle and the road. These unknown tire forces $F^\theta$ are learned as functions of the state and control inputs.

Modeling the dynamic interaction between the tires and the uncertain road surface is crucial for accurately controlling a vehicle at the limits of handling. To do so, we incorporate into our neural SDE model a version of the neural-ExpTanh tire model [15], a physics-informed neural tire model that captures the nonlinearities and saturation effects of tire forces, and that has shown to better predict tire forces than previous models used in the literature. We refer to Appendix A.1 for the derivations of $\mathrm{M}^\theta(\cdot, \cdot)$ and the neural tire force models used in the experiments.

**Model predictive control (MPC) for autonomous drifting.** The MPC tracks a reference trajectory under actuator and actuator rate constraints by solving at each time $t$ the optimization problem

$$\underset{\bar{u}_{0:H}}{\mathrm{minimize}} \quad \mathbb{E}_{\bar{x}_{1:H+1}}\Big[ \sum_{k=1}^{H} Q_\beta(\bar{\beta}_k - \beta_{\mathrm{ref},k})^2 + Q_e\bar{e}_k^2 + Q_\phi\bar{\Delta\phi}_k^2 + Q_{\dot{\delta}}\dot{\bar{\delta}}_k^2 + Q_{\dot{\tau}}\dot{\bar{\tau}}_k^{\mathrm{e}2} \Big] \qquad (9a)$$

$$\text{subject to} \quad \bar{x}_{k+1} = \mathrm{SDESolve}(\bar{x}_k, \bar{u}_k; \theta_{\mathrm{best}}) \ \forall k = 0, ..., H, \bar{x}_0 = x_t, \ \bar{u}_{0:H} \in \mathcal{U}, \dot{\bar{u}}_{0:H} \in \bar{\mathcal{U}} \quad (9b)$$

where $x_k$ is the state at timestep $k$, $\bar{x}_{1:H+1}$ denotes the state trajectory over the prediction horizon of length $H + 1$, and $\bar{u}_{0:H}$ are the corresponding control signals. SDESolve is any differentiable SDE integration scheme, in our case a simple Euler–Maruyama method, parameterized using the best parameters $\theta_{\mathrm{best}}$ found by Algorithm 2. The states $\bar{x}_{1:H+1}$ are thus random variables, and the expectation in (9a) is evaluated using Monte Carlo. We use 2 particles in our experiments. The system is actuated by $u_t = \bar{u}_0^\star$, where $\bar{u}_0^\star$ is the optimal solution to (9) constrained with $\bar{x}_0 = x_t$.

## 4 Results

We validate the proposed framework on two vehicles in scenarios with different tires, operating gears, and road conditions. First, we verify the capabilities of the conditional diffusion model to capture the

complexity of the unlabelled dataset and adapt its predictions online (Section 4.1). Second, we show that the model can adapt to different tires (Section 4.2). Third, we demonstrate the high performance of the method in diverse scenarios (Section 4.3). Finally, using a small amount of data collected on wet surfaces, we show that the framework enables drifting at the limits of handling in heavy rain (Section 4.4). We provide further details on the experiments in Appendix A.

**Experimental vehicles.** We deploy the approach on a Toyota Supra and a Lexus LC500, as shown in Figure 1. The Supra is modified with a more powerful engine and more responsive actuators, whereas the Lexus is kept with factory settings, making it a particularly challenging platform for autonomous drifting. The two vehicles' large differences in dynamics responses make them ideal platforms for evaluating the robustness and generalization capabilities of our approach. For both vehicles, we use onboard vehicle state estimation using a GPS and IMU, and we use the CPU of a ruggedized PC to run the diffusion model inference at 2Hz and compute control inputs using MPC at 200Hz.

**Training dataset.** We train the diffusion model on a total of 84 manual and autonomous driving and drifting trajectories from the two vehicles. The duration of each trajectory is between 10 and 90 seconds. It consists of 5 manual driving trajectories pushing the car to the limits of handling, whereas the remaining trajectories are autonomous drifting experiments comprising of failed and successful attempts. The dataset includes driving data in different gears (affecting the effectiveness of the throttle input) and using tires with different physical properties (affecting the vehicle's dynamics and steering input effectiveness). The composition of the dataset is summarized in Figure 3.

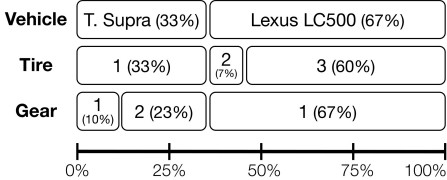

Figure 3: Dataset for model training.

**Baselines.** We compare with neural SDE dynamics models (referred to as BaseSDE or Expert depending on context) trained on specific vehicle-tire-gear subsets of the dataset in Figure 3. Each baseline is trained with the loss function in (3) and a regularization term $\mathcal{R}(\theta)$ encoding prior knowledge about the parameters $(m, a, b, I_z, I_w, R)$. The resulting models are Expert since they are optimized for specific scenarios, but they may perform poorly when deployed in different conditions.

### 4.1 Multimodality and conditioning capabilities of the diffusion model

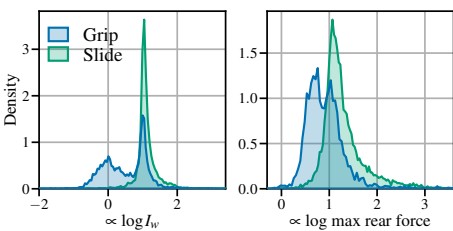

Figure 4: Parameter distribution predicted by the diffusion model, when conditioned on gripping and sliding trajectories.

We condition the model on two trajectories from the Lexus vehicle dataset (see Figure 3), where the vehicle is either accelerating in a straight line (gripping) or drifting (sliding). In Figure 4, we report two predicted parameters $(I_w^\theta, c_1^\theta)$ corresponding to the rear wheel inertia and the maximum total force that can be generated by the rear tires; see Appendix A.1. By conditioning on the straight-line (grip) trajectory, the model returns a multimodal parameter distribution due to the lack of information about tire friction properties, as tire forces are not saturated. Interestingly, by conditioning on a trajectory where the vehicle is sliding and tire forces are saturated, the model predicts a tight unimodal distribution of the parameters. The left mode of the two parameters from the gripping phase has collapsed due to sufficient information to infer vehicle properties. This example shows that the proposed conditional diffusion model captures multimodal parameter distributions and adapts its predictions based on the information contained in the trajectory.

### 4.2 Online adaptation to different tires

We further highlight the generalization capability of the proposed method by studying the closed-loop tracking performance of the Lexus when operating with various tires. We report tracking performance on two reference trajectories in Table 1. The controller using the Expert model accurately tracks the reference trajectory. In contrast, using the baseline trained only on type 3 tire data is insufficient to track the

Table 1: Tracking error: Lexus with tires type 2.

| RMSE | Donut | | Figure-8 | |
|---|---|---|---|---|
| | $e$ (m) | $\beta$ (deg) | $e$ (m) | $\beta$ (deg) |
| Expert (Tires 2) | 0.35 | 4.08 | 0.32 | 5.17 |
| BaseSDE (Tires 3) | spin | spin | spin | spin |
| BaseSDE (Tires 2 & 3) | 0.51 | 4.52 | 1.39 | 13.63 |
| Diffusion (Tires 2 & 3) | 0.31 | 4.19 | 0.56 | 5.38 |

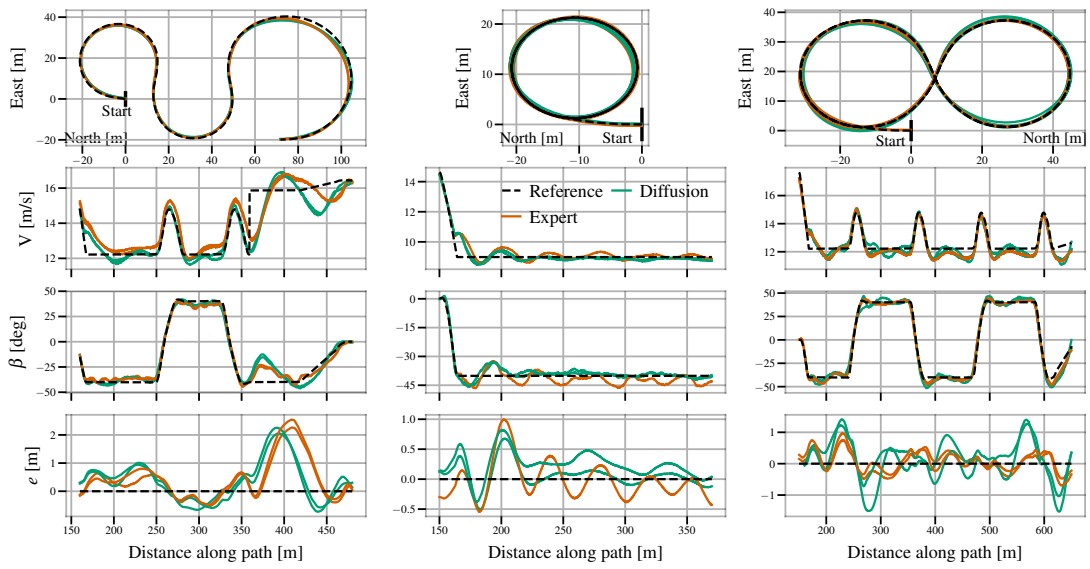

Figure 5: Drifting the Supra: performance comparison between the Expert and Diffusion models.

reference, and leads to spinning out. This baseline fails to initiate the drift, due to differences in the cornering stiffness between the two tires leading to a poor prediction of how fast the vehicle will saturate the rear tires. The BaseSDE model trained on data with type 2 and type 3 tires can drift with a higher tracking error. On the donut trajectory, this baseline drifts on a circle with a larger radius due to a tire force modeling mismatch. Finally, the diffusion model quickly infers the tire properties, which results in adaptive tracking performance that matches Expert performance.

### 4.3 Drifting performance in different scenarios

We evaluate the framework on a range of scenarios, including reference trajectories that are not in the dataset, and compare its performance with an Expert trained on the relevant subset of the dataset.

**Drifting results on the Toyota Supra.**
We report performance in tracking different reference trajectories in Figure 5 and Table 2. The diffusion model enables drifting maneuvers with tracking perfor-

Table 2: Tracking error on the Toyota Supra.

| RMSE | Slalom (gear 2) | | Donut (gear 1) | | Figure-8 (gear 2) | |
|---|---|---|---|---|---|---|
| | $e$ (m) | $\beta$ (deg) | $e$ (m) | $\beta$ (deg) | $e$ (m) | $\beta$ (deg) |
| Expert | 0.74 | 5.03 | 0.31 | 3.32 | 0.32 | 3.40 |
| Diffusion | 0.72 | 6.52 | 0.19 | 2.26 | 0.57 | 4.34 |

mance that is comparable to expert models, while simultaneously having the advantage of being trained on an unstructured dataset. This demonstrates the model's ability to adapt at test time to the specific vehicle setting and road condition based on the online observation. Interestingly, the framework succeeds in accurately tracking the Slalom trajectory (first column of Figure 5), which is not part of any maneuvers in the training dataset. Tracking this trajectory requires the vehicle to operate outside of the training data distribution, given the rapid changes between circles of different radii while accelerating during the last transition of the trajectory. We speculate that this ability to generalize results from the prior physics knowledge encoded in the neural SDE vehicle model.

**Drifting results on the Lexus LC500.**
Tracking results in Figure 6 and Table 3 show that using the diffusion model enables accurate tracking with performance comparable to expert models. Again, the

Table 3: Tracking error: Lexus with type 3 tires.

| RMSE | Donut (gear 2) | | Donut (gear 1) | | Figure-8 (gear 1) | |
|---|---|---|---|---|---|---|
| | $e$ (m) | $\beta$ (deg) | $e$ (m) | $\beta$ (deg) | $e$ (m) | $\beta$ (deg) |
| Expert | 0.48 | 3.44 | 0.38 | 6.39 | 0.32 | 3.27 |
| Diffusion | 0.82 | 4.70 | 0.29 | 4.48 | 0.34 | 3.79 |

proposed method is capable of performing a donut trajectory in second gear, although no second-gear trajectory from the Lexus is in the dataset. Moreover, only unsuccessful Figure-8 trajectories on the Lexus are in the dataset, yet both methods using the expert and diffusion models succeed in tracking this trajectory, thanks to the physics structure encoded in the neural SDE model.

### 4.4 Drifting in low-friction conditions using limited data

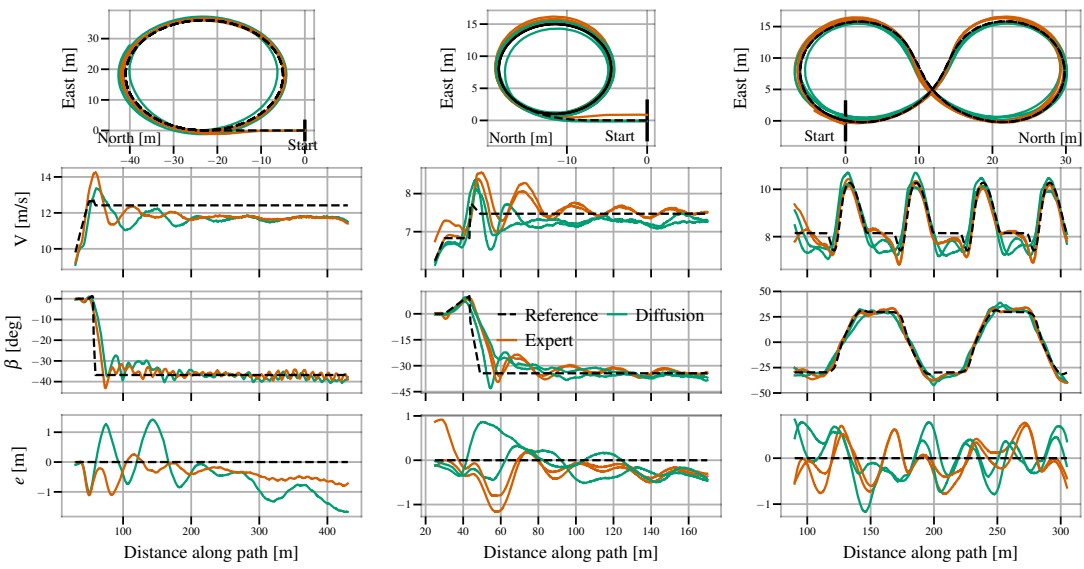

Figure 6: Drifting the Lexus: performance comparison between the Expert and Diffusion models.

Lastly, we augment the dataset with 3 manual drifting trajectories collected in the rain and 4 autonomous drifting donut trajectories on a wet surface with failed and successful attempts on the Lexus vehicle. Then, we retrain the diffusion model and report the performance of the resulting controller deployed in heavy rain in Figure 7. The proposed approach is capable of drifting with only $1.47$ m as the lateral RMS error ($e$) and $4.79$ deg as the RMS slip angle ($\beta$) error.

Drifting in such rainy conditions is particularly challenging. Indeed, accurate friction modeling is critical to successfully initiating the drift without spinning out, especially using a commercial vehicle such as the Lexus. Stabilizing the vehicle in rainy conditions is particularly difficult due to the increased effect of small friction variations on the handling characteristics of the vehicle, and because friction parameters vary over space due to the terrain drying unevenly over time. These results indicate that a single generative model, trained on a majority of data collected on high-friction surfaces, has the potential to enable reliable autonomous driving at the limits in both high and low-friction conditions.

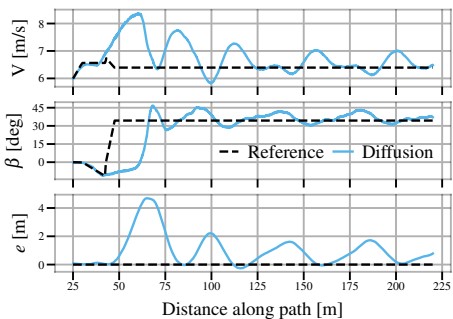

Figure 7: Tracking performance in heavy rain on a donut trajectory when drifting the Lexus.

## 5 Conclusions

We propose a physics-informed generative vehicle model for autonomous driving at the limits of handling. By decoupling model inference and control, this hierarchical approach combines the expressiveness of a diffusion model with the high-rate replanning and reliability of model predictive control. Through extensive autonomous drifting experiments on a Toyota Supra and Lexus LC500, we demonstrate that a single conditional diffusion model, trained on unlabelled trajectories from both vehicles operating in various conditions, can enable adaptive, robust, and real-time autonomous driving at the limits of handling.

**Limitations and future work.** Although the diffusion model predicts a multimodal distribution of parameters of the neural SDE vehicle dynamics model, the current model predictive controller only uses one predicted parameter set for control at a time. Fully reasoning over predicted distribution using risk-sensitive algorithms would potentially lead to additional robustness and inform data collection via active exploration to best reduce uncertainties online. Finally, while only validated on drifting tasks, the generality of the proposed hierarchical method and the experiments indicate that the approach could potentially be used in other autonomous driving and robotics applications.

## Acknowledgments

We would like to thank the platform research team at Toyota Research Institute for their support with the test platforms and experiments. Special acknowledgment to Phung Nguyen and Steven Goldine for facilitating the experiments and making it possible to validate our framework under various conditions, and Jenna Lee for facilitating data logging and processing. We would also like to thank Yusei Sakamoto at Toyota Motor Corporation for the tremendous assistance in setting up our framework on their test platform, and creating an environment to validate our approach on wet surfaces and rainy conditions.

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

# A   Appendix

This section provides additional details on the neural SDE vehicle model, the experimental vehicles and training dataset, the expert models used as baselines, the diffusion model training, and the model predictive control formulation.

We implement all the numerical experiments (training the models and the gradient-based model predictive control solver) using the Python library JAX [92] to take advantage of its automatic differentiation and just-in-time compilation features. We use Python 3.8.5 for the experiments and train all our models on a laptop computer with an Intel® Xeon(R) W-11855M CPU (base frequency 3.30GHz), 12 cores, 32 GB of RAM, and a GeForce RTX 2060, TU10.

## A.1   Physics-inspired neural SDE vehicle model

We employ the commonly used single-track model [90, 16, 91, 18, 17] as a foundation to describe the nonlinear dynamics of the vehicle. The vehicle position is expressed in a curvilinear coordinate system relative to a reference trajectory [14, 19, 12], as shown in Figure 8. Specifically, the position coordinate is described by the distance $s$ along the path, the relative heading $\Delta\phi$ with respect to a planned course $\phi_{\text{ref}}$, and the lateral deviation $e$ from the path. For simplicity, we assume only the steering and throttle are used for autonomous drifting and do not include brakes in the control inputs and dynamics. The proposed neural SDE model is given by

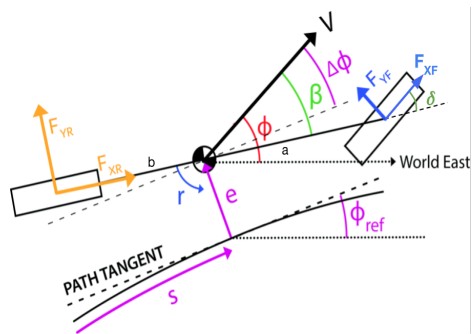

Figure 8: Single-track model of a vehicle on a reference path.

$$
\mathrm{d}
\begin{bmatrix}
r \\[4pt]
V \\[4pt]
\beta \\[4pt]
\omega_r
\end{bmatrix}
=
\begin{bmatrix}
\dfrac{a^\theta F_{yf}^\theta \cos(\delta) + a^\theta F_{xf}^\theta \sin(\delta) - b^\theta F_{yr}^\theta}{I_z^\theta} \\[12pt]
\dfrac{-F_{yf}^\theta \sin(\delta-\beta) + F_{xf}^\theta \cos(\delta-\beta) + F_{yr}^\theta \sin(\beta) + F_{xr}^\theta \cos(\beta)}{m^\theta} \\[12pt]
\dfrac{F_{yf}^\theta \cos(\delta-\beta) + F_{xf}^\theta \sin(\delta-\beta) + F_{yr}^\theta \cos\beta - F_{xr}^\theta \sin\beta}{m^\theta V} - r \\[12pt]
\dfrac{GE^\theta(\tau^{\mathrm e}) - F_{xr}^\theta R^\theta}{I_w^\theta}
\end{bmatrix}
+ \Sigma^\theta(x, u)\mathrm{d}W, \qquad (10)
$$

where vehicle-specific parameters such as the mass $m^\theta$, yaw moment of inertia $I_z^\theta$, rotational inertia of the drivetrain $I_w^\theta$, the tire radius $R^\theta$, and the distances from the center of gravity to the front and rear axles $a^\theta$ and $b^\theta$ are included in the neural SDE parameters $\theta$ to learn. The control input $u = [\delta, \tau^{\mathrm e}]$ is the steering angle and engine torque, respectively. $E^\theta(\tau^{\mathrm e})$ is a parameterized polynomial function that maps the engine torque to the wheel torque through the gear ratio $G$. The state $x = [r, V, \beta, \omega_r, e, \Delta\phi, s]$ includes the yaw rate $r$, velocity $V$, sideslip angle $\beta$, rear wheelspeed $\omega_r$, lateral error $e$, and angular deviation $\Delta\phi$. The evolution of the path-dependent variables $e$, $\Delta\phi$, and $s$ are well described by basic kinematics

$$
\mathrm{d}e = V \sin(\Delta\phi)\mathrm{d}t, \qquad (11)
$$

$$
\mathrm{d}s = \frac{V \cos(\Delta\phi)}{1 - e\kappa_{\text{ref}}(s)}\mathrm{d}t, \qquad (12)
$$

$$
\mathrm{d}(\Delta\phi) = (\dot\beta + r - \kappa_{\text{ref}}(s)\dot s)\mathrm{d}t, \qquad (13)
$$

where $\kappa_{\text{ref}}(s)$ is the curvature of the reference path, and we use $\dot\beta$ and $\dot s$ as an abuse of notation for the drift terms of $\beta$ and $s$, respectively. Lastly, the unknown lateral and longitudinal tire forces $F_{xf}^\theta$, $F_{yf}^\theta$, $F_{xr}^\theta$, and $F_{yr}^\theta$ are parametrized and learned as functions of the state and control inputs.

We propose to incorporate into our neural SDE model a version of the recently proposed neural-ExpTanh [15] tire mode parameterized by

$$
F_{yf}^\theta = \mathrm{ExpTanh}^\theta(\tan(\alpha_f); \mathrm{feat}_1), \ F_{\mathrm{tot}}^\theta = \mathrm{ExpTanh}^\theta(\tan^2(\alpha_r) + c_0^\theta \sigma_r^2; \mathrm{feat}_2),
$$

$$
\begin{bmatrix} F_{yr}^\theta \\ F_{xr}^\theta \end{bmatrix} = \frac{\mathrm{NN}_0^\theta(\alpha_r, \sigma_r)}{\|\mathrm{NN}_0^\theta(\alpha_r, \sigma_r)\|} F_{\mathrm{tot}}^\theta, \ \sigma_r = \frac{R^\theta \omega_r - V\cos\beta}{V\cos\beta},
$$

$$
\alpha_f = \mathrm{atan} \frac{V\sin\beta + a^\theta r}{V\cos\beta} - \delta, \ \alpha_r = \mathrm{atan}\frac{V\sin\beta - b^\theta r}{V\cos\beta},
$$

(14)

where $\mathrm{ExpTanh}^\theta(z; \mathrm{feat}) := c_1^\theta + c_2^\theta e^{-c_3^\theta |z|} \tanh\left(c_4^\theta(z - c_5^\theta)\right)$ is such that $(c_i^\theta)_{i=1}^5 = \mathrm{NN}^\theta(\mathrm{feat})$ is the output of a neural network with input feat and satisfying $c_3^\theta, c_4^\theta \geq 0$ (enforced via an exponential function on the last two outputs of the neural network). Besides, $\sigma_r$ is the slip ratio, $\alpha_f$ and $\alpha_r$ are the slip angles for the front and rear tires, $\mathrm{feat}_1 = [r, V, \beta]$ and $\mathrm{feat}_2 = [r, V, \beta, \omega_r]$ are features for the two neural-ExpTanh models, and $c_0^\theta$ and $\mathrm{NN}_0^\theta$ are learned to approximate the coupled effect between the longitudinal and lateral tire dynamics when the vehicle is sliding and accelerating at the same time. We note that our model has four neural networks: $\mathrm{NN}_0^\theta$ for the coupled effect, $\mathrm{NN}_1^\theta$ and $\mathrm{NN}_2^\theta$ for front and rear neural-ExpTanh tire models, and $\Sigma^\theta$ for the diffusion term. Lastly, we emphasize that although the proposed neural model assumes rear-wheel drive vehicles, it can be straightforwardly extended to other drive configurations.

**Modeling details.** We use a diagonal matrix to represent the noise scale $\Sigma^\theta$. This design choice greatly reduces the computation of the losses (2), (3), and (4) at the cost of possibly limiting the expressivity of the model by neglecting correlations between states. However, the experiments show that the model can still capture the complex dynamics of the vehicle and enable reliable performance on diverse drifting maneuvers. The vehicle parameters $m^\theta, I_z^\theta, I_w^\theta, R^\theta, a^\theta$, and $b^\theta$ are learnable scalar values optimized during training. We refer to Appendix A.3 for prior knowledge enforced on these vehicle parameters during the training of expert models, but not during the diffusion model training. The engine torque function $E^\theta(\tau^{\mathrm{e}})$ is a linear function with learnable parameters. The parameter $c_0^\theta$ is a learnable scalar value, and the neural networks $\mathrm{NN}_0^\theta, \mathrm{NN}_1^\theta$, and $\mathrm{NN}_2^\theta$ are feedforward neural networks with two hidden layers of 6 neurons each and $\tanh$ activation functions. We do not perform any preprocessing on the training dataset and learn from raw and noisy vehicle trajectories.

### A.2 Experimental vehicles and training dataset

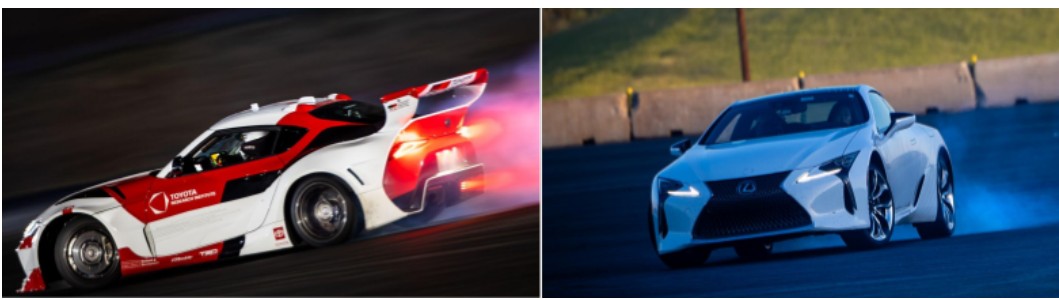

Figure 9: Experimental vehicles used in the study. Left: Toyota Supra. Right: Lexus LC500.

**Experimental vehicles.** We deployed our framework on a Toyota Supra and a Lexus LC500, illustrated in Figure 9. The Supra is a prototype vehicle that has been heavily modified to become an autonomous drifting platform, while the Lexus is a commercial 2019 LC500 with its powertrain, drivetrain, and suspension unmodified from the dealership. The Supra has been modified with a 3L inline-six engine capable of outputting 380hp. The engine has been outfitted with an upgraded turbocharger that can provide an additional 300hp, for a total of 680hp. The steering system is outfitted with both hydraulic assist and electric power-assisted steering, providing high-performance steer-by-wire abilities with up to 56.7Nm of torque. The vehicle has also been modified to provide

brake-by-wire and throttle-by-wire capabilities. These modifications make the platform extremely suitable for (autonomous) Formula drift, as they allow for precise and fast control of the vehicle's dynamics. The Lexus, on the other hand, has lower-performance actuators and is not designed to be used as a drifting vehicle. Ultimately, these two platforms have completely different dynamics, which makes them ideal for evaluating the robustness and generalization capabilities of our approach.

We use an Oxford Technical Systems (OxTS) RT4003 v2 RTK-GPS/IMU system for localization and vehicle state estimation for both vehicles. The MPC controller is implemented on an Intel Xeon E-2278GE (base frequency 3.30GHz) CPU Linux computer mounted on board the vehicles. The computer communicates with a low-level PID controller implemented on a dSpace MicroAutoBoxII (DS1401) to track desired steering angle $\delta$ and engine torque $\tau^e$. The MicroAutoBoxII receives commands from the MPC controller via UDP and sends actuator commands to the original equipment manufacturer's steering and engine electronic control units. All data are synchronized and recorded at a frequency of 62.5Hz on the Lexus and 100Hz on the Supra.

**Training dataset.**     We build a dataset of manual and autonomous driving and drifting trajectories on a closed circuit from both vehicles. The dataset contains a total of 84 trajectories, each trajectory with a duration between 10 and 90 seconds. It has 5 trajectories collected from manual driving with the intent of pushing the car to the limits of handling without any specific path-tracking maneuvers planned. The remaining trajectories are from autonomous drifting experiments with 28 from the Supra and the rest from the Lexus. The supra dataset contains failed and successful attempts at performing donut maneuvers in first gear and Figure-8 maneuvers in second gear. The Lexus dataset contains attempts at performing donut maneuvers in first gear and very limited (all failed) attempts at performing "Figure-8" maneuvers in first gear. No second-gear drifting trajectories were provided in the training dataset for the Lexus. Additionally, 7% of the trajectories from the Lexus were collected with Tire 2, as opposed to Tire 3 used in the rest of the dataset. A notable difference between the two sets of tires is their cornering stiffness, which makes drift initiation strategies and tire dynamics different between the two sets of tires.

**Training dataset for drifting on heavy rain.**     We augmented the above training dataset with 7 additional drifting trajectories collected on a wet track with a vehicle identical to the Lexus LC500. 3 of the trajectories were manually collected, while the remaining 4 were collected autonomously. The autonomous trajectories were collected using second-gear drifting maneuvers only on a donut trajectory. Despite the limited number of trajectories and restriction to second-gear drifting, we show in Section 4.4 that the diffusion model trained on this dataset generalizes to drifting in heavy rain on a first-gear donut trajectory.

### A.3   Expert models training

Table 4: Prior parameters for the expert models.

| Vehicle | $m$ (kg) | $I_z$ (kg·m$^2$) | $I_w$ (kg·m$^2$) | $R$ (m) | $a$ (m) | $b$ (m) |
|---|---|---|---|---|---|---|
| Toyota Supra | 2048 | 3675 | 6 | 0.368 | 1.345 | 1.522 |
| Lexus LC500 | 1476 | 2241 | 6 | 0.323 | 1.239 | 1.209 |

We recall that the expert models in the experiments are neural SDE models trained on specific vehicle-tire subsets of the dataset. Given a subset $\mathcal{T}_{\exp}$ of the training dataset $\mathcal{T}$, we train the expert models by minimizing the negative log-likelihood $\bar{\mathcal{J}}_{\text{traj}}(\theta, \mathcal{T}_{\exp})$ defined in (3) with respect to the neural SDE parameters $\theta$ defined in Appendix A.1. The regularization term $\mathcal{R}(\theta)$ encodes a Gaussian prior on the vehicle parameters $m^\theta$, $I_z^\theta$, $I_w^\theta$, $R^\theta$, $a^\theta$, and $b^\theta$. For each parameter, we use a Gaussian prior with a mean equal to the known (estimated) parameter value for the specific vehicle and a standard deviation of 1. The resulting $\mathcal{R}(\theta)$ is given by

$$\mathcal{R}(\theta) = (m^\theta - m)^2 + (I_z^\theta - I_z)^2 + (I_w^\theta - I_w)^2 + (R^\theta - R)^2 + (a^\theta - a)^2 + (b^\theta - b)^2, \quad (15)$$

where the parameters values $m$, $I_z$, $I_w$, $R$, $a$, and $b$ for each vehicle are provided in Table 4. The expert models are trained using Adam optimizer [93] with a learning rate of $10^{-3}$ and a batch size of 64. We use $\lambda_{\text{traj}} = 10^{-4}$ for the regularization term in the loss function (3). During training, we discretize the sum in the expression of $\mathcal{J}_{\text{nll}}$ (see (2) for the expression and (3) for the loss function) into a sum of $N_{\text{f}} = 20$ discrete time steps $t_0, \ldots, t_{N_{\text{f}}}$, where $t_0 = t$, $\text{T}_{\text{f}} = t_N - t_0$, and $t_i = t_{i-1} + \Delta t_i$

for $i = 1, \ldots, N_{\mathrm{f}}$. We set $\Delta t_i = \mathbb{U}_{(1,6)} \Delta t$ where $\Delta t$ is typically $0.01$ for trajectories on the Supra and $0.016$ for trajectories on the Lexus. We randomize the time steps $\Delta t_i$ during training to improve the model's generalization when evaluated with an integration scheme that uses stepsizes other than the training stepsizes. This is typically the case when using the model in a model predictive controller; see Appendix A.5. We note that the model learns to predict state-action sequences with varying lengths $\mathrm{T}_{\mathrm{f}}$ between $0.4$ seconds and $1.96$ seconds. Lastly, we use 5 particles of the neural SDE to compute the expectation in the expression of $\mathcal{J}_{\mathrm{nll}}$.

### A.4 Diffusion model training and online sampling

**Initial estimate of the neural SDE parameters.** We train the initial estimate $\theta^{\mathrm{loc}}$ of the neural SDE parameters in a similar manner to the expert models in Appendix A.3. The main difference is that we use the full training dataset $\mathcal{T}$ instead of the vehicle-specific subset $\mathcal{T}_{\mathrm{exp}}$, and thus we do not enforce any prior knowledge on the vehicle parameters as in (15). Specifically, the regularization term $\mathcal{R}(\theta)$ is now enforcing an uninformative Gaussian prior with mean 0 and standard deviation 1 on all the neural SDE parameters $\theta$ as follows

$$\mathcal{R}(\theta) = \sum\nolimits_{i=1}^{N_\theta} (\theta_i)^2, \tag{16}$$

where $N_\theta$ is the number of parameters in $\theta$, and $\theta$ in defined as in Appendix A.3.

**Parameter dataset generation via local optimization.** The hyperparameters that define this step of the diffusion model training are the regularization parameter $\lambda_{\mathrm{loc}}$, the time window $W$, the history length $\mathrm{T}_{\mathrm{p}}$, the future trajectory length $\mathrm{T}_{\mathrm{f}}$, and the number of sequences $\tau_{\mathrm{T}_{\mathrm{p}}:t_k:\mathrm{T}_{\mathrm{f}}}$ used to compute the expectation in (4). We set $\lambda_{\mathrm{loc}} = 10^{-3}$, $\mathbb{W} = 10$ seconds, and the number of sequences $\tau_{\mathrm{T}_{\mathrm{p}}:t_k:\mathrm{T}_{\mathrm{f}}}$ to 5. In a similar manner to how we chose the time steps in the training of the expert models (see Appendix A.3), we discretize $\tau_{\mathrm{T}_{\mathrm{p}}:t}$ and $\tau_{\mathrm{T}_{\mathrm{p}}:t_k}$ into $N_{\mathrm{p}} = 10$ discrete time steps with the same randomization of the step size $\Delta t_i$ as in Appendix A.3. Thus, the maximum length of the history sequence is $\mathrm{T}_{\mathrm{p}} = 0.96$ seconds. Besides, we keep the future trajectory length $\mathrm{T}_{\mathrm{f}}$ to be the same as in the training of the expert models, i.e., with $N_{\mathrm{f}} = 20$ discrete time steps and a maximum length of $1.96$ seconds. We use 5 particles of the neural SDE to compute the expectation in the expression of $\mathcal{J}_{\mathrm{nll}}$.

We optimize the neural SDE parameters $\theta$ of the loss function (4) using gradient descent with Nesterov acceleration and an adaptive learning rate through Armijo line search. We set the maximum number of iterations to 1000 and the initial guess for the learning rate and neural SDE parameters to be respectively $0.01$ and $\theta^{\mathrm{loc}}$.

**Diffusion model training.** We follow the procedure described in [7] to train all our diffusion models. The model $\epsilon_\psi$ defining the generative process is represented as a standard feedforward neural network with three hidden layers of 256 neurons each. We use a sinusoidal positional encoding of the diffusion step $k$ (see (5)) as an input to the neural network defining $\epsilon_\psi$, instead of using $k$ directly as the input. The encoding is done by scaling the diffusion step and concatenating its sine and cosine to the input of a feedforward neural network with two hidden layers of 32 and 16 neurons each and swish as the activation function. We use $K = 1000$ denoising steps, and a linear noise schedule $\beta_i \in (0, 1)$, where $\beta_i = 0.0001 + 0.02i/K$ for $i = 0, \ldots, K$. We use Adam optimizer with a learning rate of $10^{-4}$ and a batch size of 32 to train the diffusion model. Additionally, we perform 50 gradient updates of $\epsilon_\psi$ for each step of Algorithm 1.

**Iteratively refining the parameter dataset.** Given the history sequence $\tau_{\mathrm{T}_{\mathrm{p}}:t}$, we use the diffusion generation process defined in (7) to obtain a set of parameters $\{\theta_t^p\}_{p=0}^{100}$ conditioned on $\tau_{\mathrm{T}_{\mathrm{p}}:t}$. Then, we use the future sequence $\tau_{t:\mathrm{T}_{\mathrm{f}}}$ to select the best parameter in terms of the negative log-likelihood $\mathcal{J}_{\mathrm{nll}}(\theta_t^p, \tau_{\mathrm{T}_{\mathrm{p}}:t:\mathrm{T}_{\mathrm{f}}})$ with 5 particles of the neural SDE to compute the expectation in its expression. The obtained best parameter is then used to update $\theta^{\mathrm{loc}}$ for better initialization and regularization of the local optimization problem (4). In our experiments, we refine $\theta^{\mathrm{loc}}$ at every step of Algorithm 1 only after the initial 20000 steps of the main training loop.

**Online diffusion model inference.** In Algorithm 2, we use a sliding window to deal with the growing size of the dataset $\mathcal{T}_{\mathrm{hist}}$ and to account for changing vehicle-road properties or environment conditions. The maximum size of the sliding window $\mathcal{T}_{\mathrm{hist}}$ is typically set to 30 seconds worth of

driving data. During online sampling for model predictive control, we randomly sample the set $\mathcal{T}_{\text{gen}}$ to contain 5 history sequences $\tau_{\text{T}_{\text{p}}:t_j}$ and the set $\mathcal{T}_{\text{val}} \subseteq \mathcal{T}_{\text{hist}}$ to contain 30 sequences $\tau_{t_l:\text{T}_{\text{f}}}$ of the current history dataset $\mathcal{T}_{\text{hist}}$. Specifically, by indexing each state-action pair $x_{t_k}, u_{t_k}$ with the corresponding discrete time, we can define a discrete distribution to sample sequences $\tau_{\text{T}_{\text{p}}:t_k}$, by sampling time indexes $t_k$ and using the corresponding state-action as the endpoint of the sequence of length $\text{T}_{\text{p}}$. In our experiments, we use an exponential distribution with a higher mass on the latest time indexes in $\mathcal{T}_{\text{hist}}$ to generate $\mathcal{T}_{\text{gen}}$. Then, to generate $\mathcal{T}_{\text{val}}$, we pick the future sequences $\tau_{t_j:\text{T}_{\text{f}}}$ corresponding to the sequences of $\mathcal{T}_{\text{gen}}$ and sample the remaining validation sequences $\tau_{t_l:\text{T}_{\text{f}}}$ in the same manner as $\mathcal{T}_{\text{gen}}$. The sequences in the validation dataset are selected to be as close as possible to the latest time in the history dataset. We use the diffusion model to generate a total of 100 parameters $\{\theta_{t_k}\}_k$ conditioned on the sequences in $\mathcal{T}_{\text{gen}}$, and select the best parameter according to Algorithm 2.

### A.5 Model predictive control formulation

We use a custom proximal gradient-based solver with Nesterov acceleration and Armijo line search, inspired by the approach in [94], to optimize the MPC problem 9a–9b. The state constraints, if any, and control rate constraints are enforced using slack variables, and the proximal operator for projecting the slack variables onto the feasible set. We use a first-order approximation to compute the control rate as in $\dot{\bar{u}}_k = (\bar{u}_{k+1} - \bar{u}_k)/\Delta t_k$. On the other hand, the box constraints on the control input are simply enforced by projection onto the set at each iteration of the proximal gradient-based solver.

**Reference trajectories.** The reference trajectories for the maps are generated offline via the quasi-equilibrium strategy proposed in [19]. We start with a few waypoints in terms of the desired curvature $\kappa_{\text{ref}}$ and sideslip angle $\beta_{\text{ref}}$ as a function of path distance $s$. Then, for each point on the path, an equilibrium point is computed using the single-track bicycle model and the conditions $\kappa = \kappa_{\text{ref}}$, $\beta = \beta_{\text{ref}}$, $\dot{\phi} = \kappa_{\text{ref}}V$, and $\dot{V} = \dot{r} = 0$, yielding the fine-grained reference vehicle state $x_{\text{ref}}$. We emphasize that the model used to generate the reference trajectories differs significantly from the neural SDE model used in the MPC controller. We reduce the over-reliance on possibly infeasible reference trajectory by using a cost function that penalizes the deviation from only in the sideslip angle $\beta$ and wheel speed $\omega_r$, see (9a)-(9b).

**Lexus LC500.** We use $Q_\beta = 120$, $Q_e = 2.0$, $Q_\phi = 60.0$, $Q_{\dot{\delta}} = 5$, and $Q_{\dot{\tau}} = 10^{-6}$. The control set is given by $\mathcal{U} = [-0.52, 0.52] \times [-1, 400]$ while the control rate set is given by $\bar{\mathcal{U}} = [-0.9, 0.9] \times [-3000, 400]$. The problem is optimized over a horizon $H = 30$ with 25 short time steps of $0.05$s and 5 long time steps of $0.15$s. Thus, the total lookahead horizon amounts to 2s.

**Toyota Supra.** We use $Q_\beta = 70$, $Q_e = 3.0$, $Q_\phi = 30.0$, $Q_{\dot{\delta}} = 1$, and $Q_{\dot{\tau}} = 10^{-7}$. The control set is given by $\mathcal{U} = [-0.75, 0.75] \times [-50, 350]$ while the control rate set is given by $\bar{\mathcal{U}} = [-2, 2] \times [-3000, 2000]$. The problem is optimized over a horizon $H = 30$ with 25 short time steps of $0.05$s and 5 long time steps of $0.15$s. Thus, the total lookahead horizon amounts to 2s.

### A.6 Details on the drifting experiments

In this section, we provide additional details on the drifting experiments conducted with the Toyota Supra and Lexus LC500 vehicles. We show the full vehicle state and control evolution when drifting on the various trajectories and road conditions, and when equipped with different tires:

- Figure 10 shows the Lexus LC500 drifting on a first-gear donut trajectory with Tire 3.
- Figure 11 shows the Lexus LC500 drifting on a first-gear Figure-8 trajectory with Tire 3.
- Figure 12 shows the Lexus LC500 drifting on a second-gear donut trajectory with Tire 3.
- Figure 13 shows the Toyota Supra drifting on a first-gear donut trajectory.
- Figure 14 shows the Toyota Supra drifting on a second-gear Figure-8 trajectory.
- Figure 15 shows the Toyota Supra drifting on a second-gear slalom-like trajectory.
- Figure 16 shows the Lexus LC500 drifting on a first-gear donut trajectory with Tire 2.
- Figure 17 shows the Lexus LC500 drifting on a first-gear Figure-8 trajectory with Tire 2.
- Figure 18 shows the Lexus LC500 drifting on a first-gear donut trajectory on heavy rain.

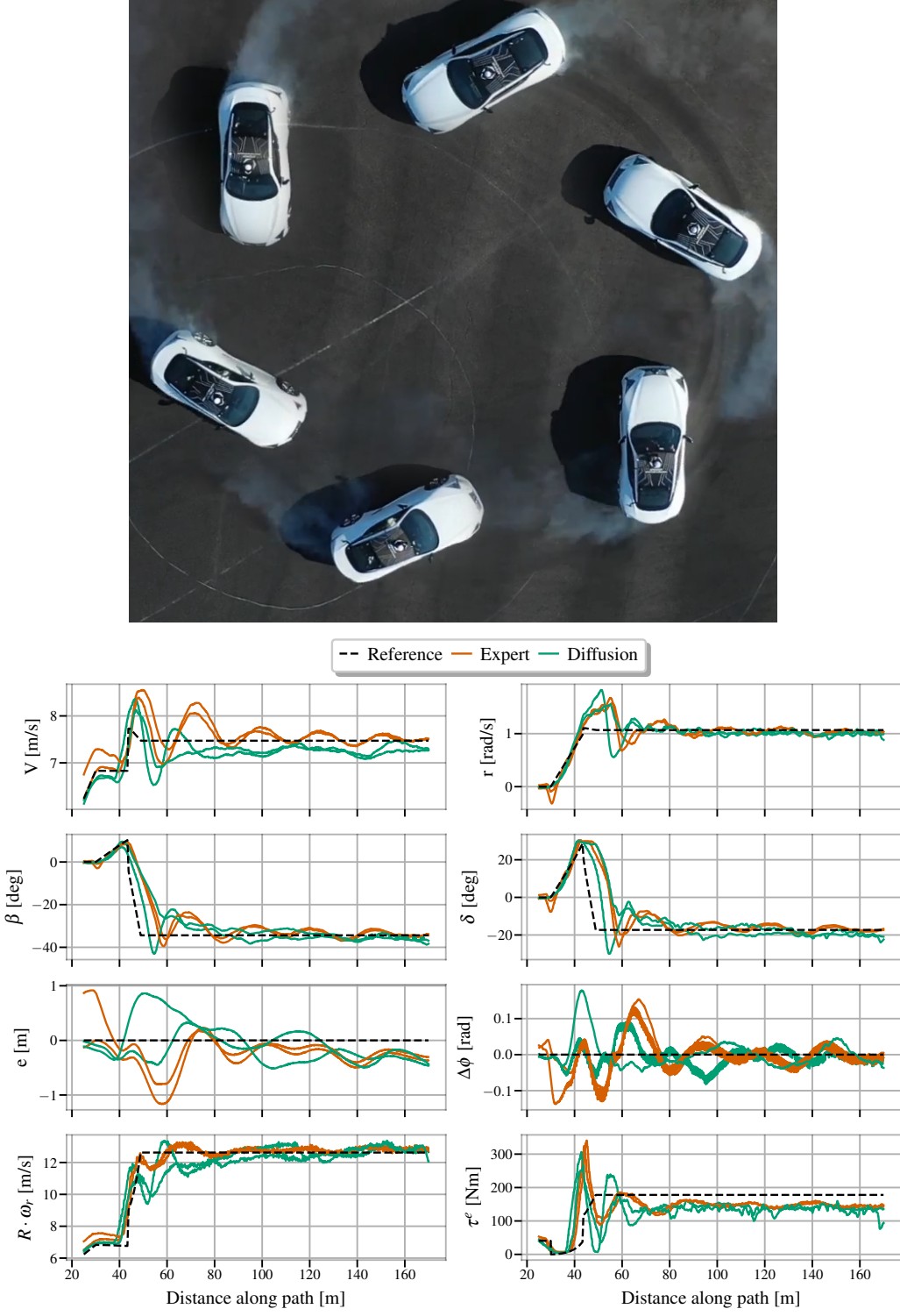

Figure 10: Lexus drifting on a first-gear donut trajectory with Tire 3.

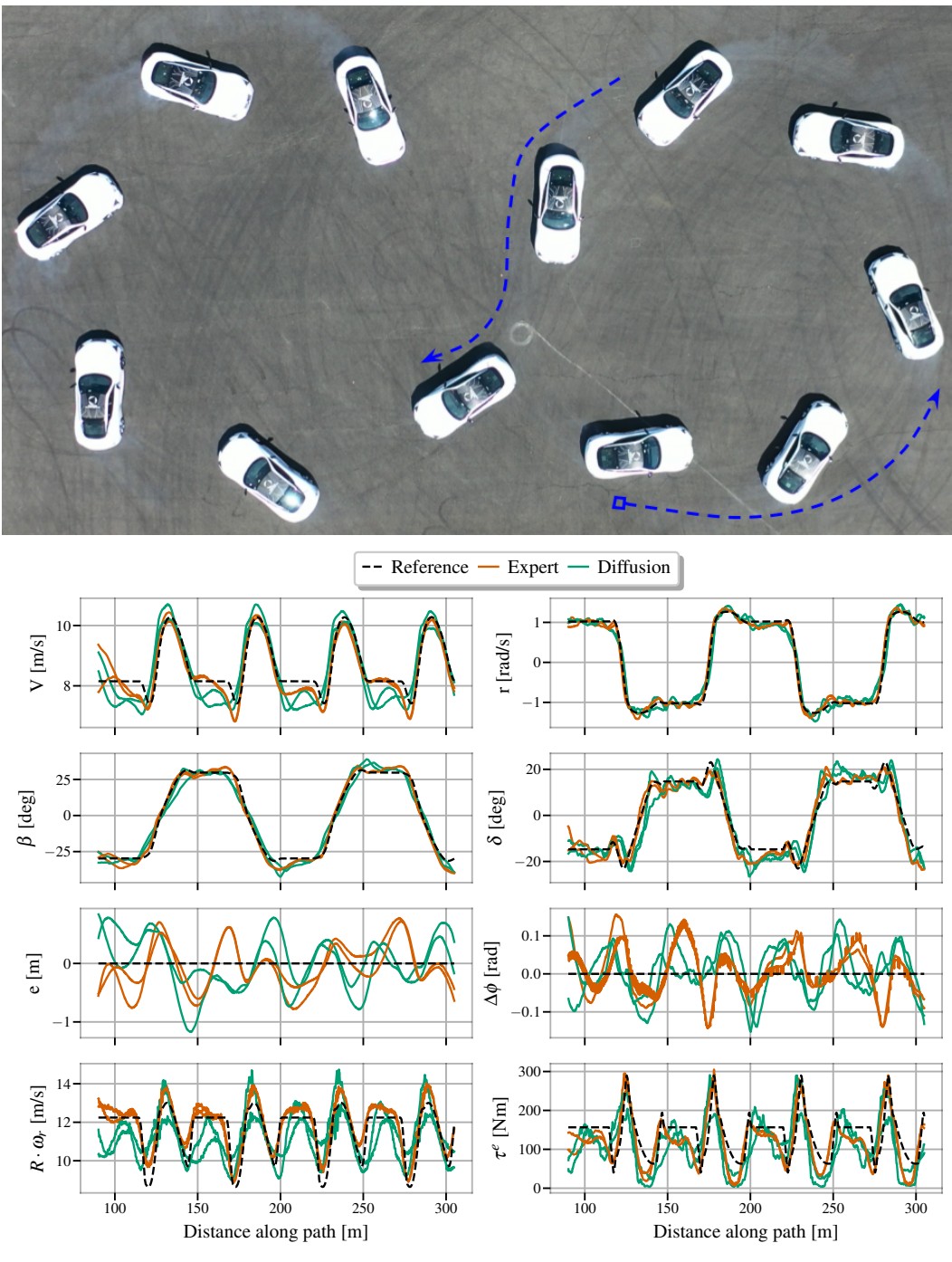

Figure 11: Lexus drifting on a first-gear Figure-8 trajectory with Tire 3.

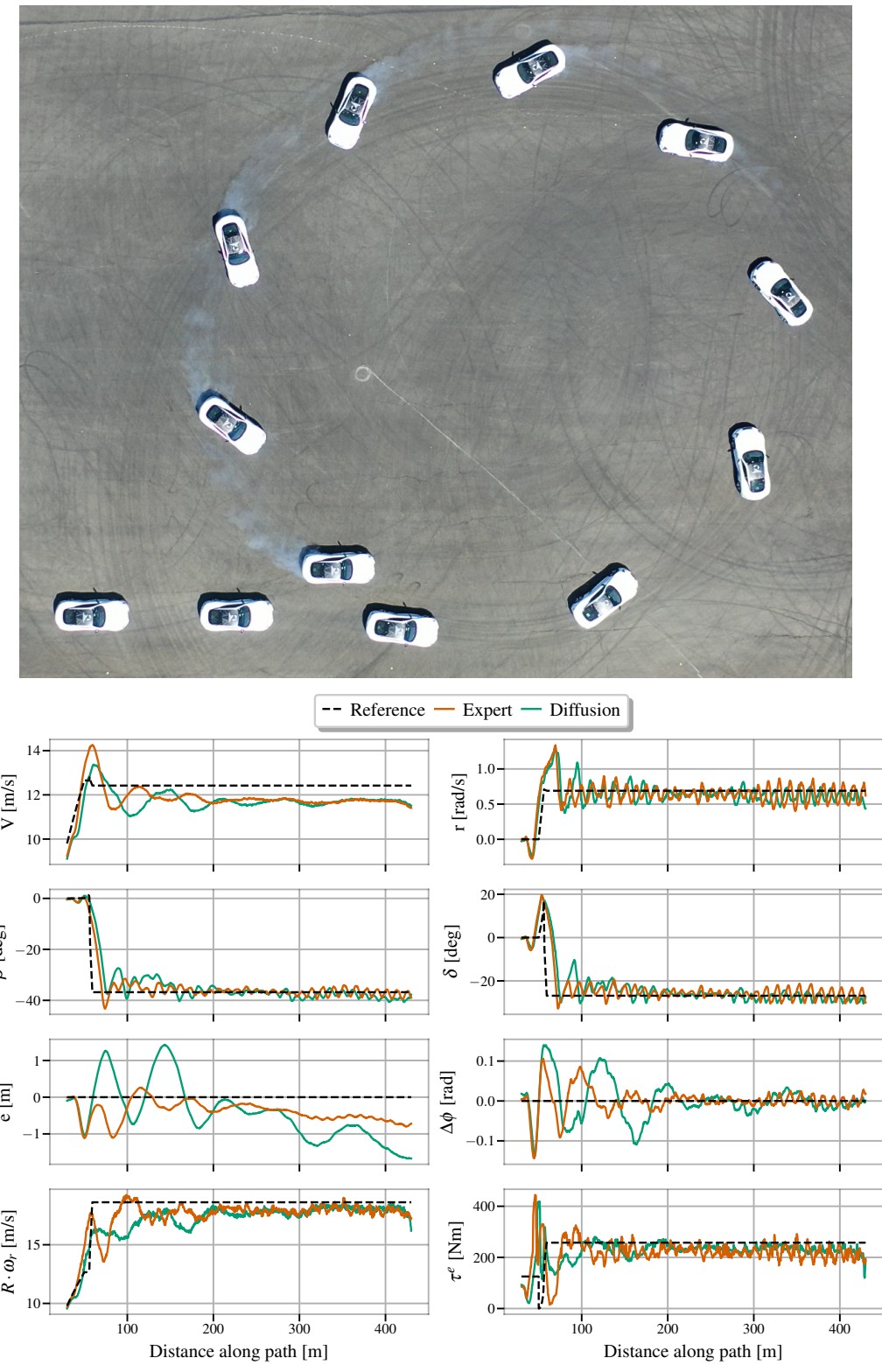

Figure 12: Lexus drifting on a second-gear donut trajectory with Tire 3.

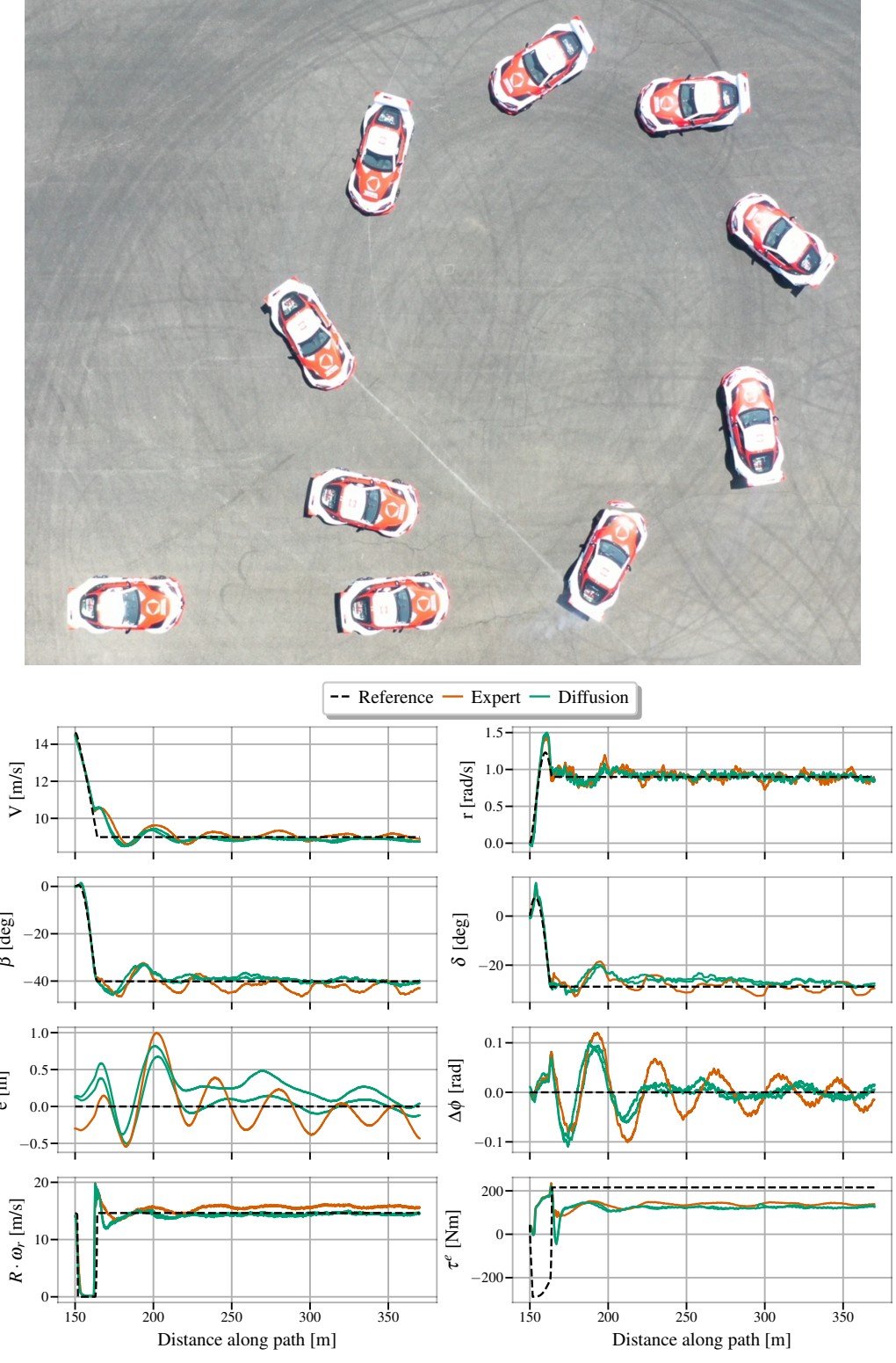

Figure 13: Toyota Supra drifting on a first-gear donut trajectory.

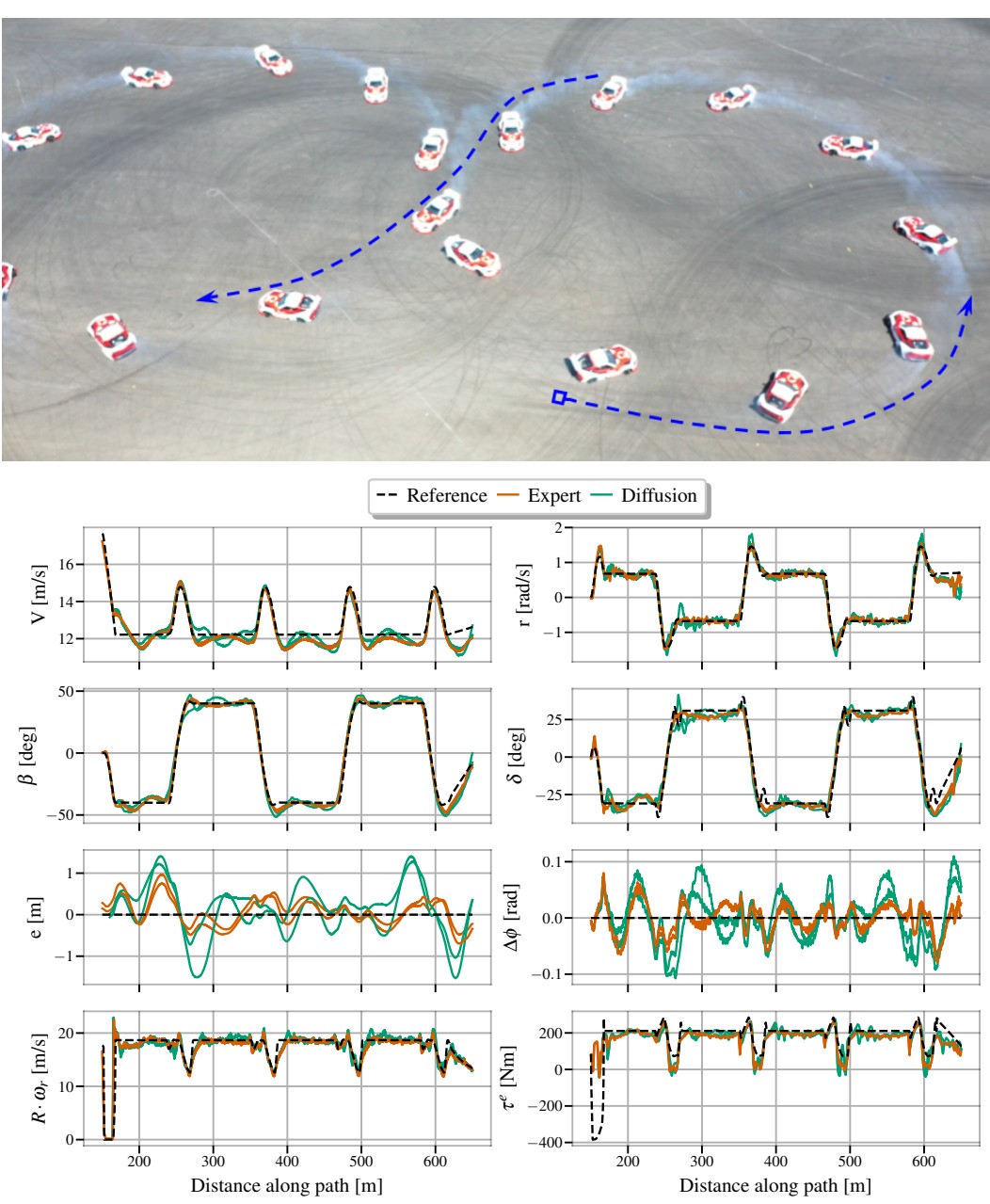

Figure 14: Toyota Supra drifting on a second-gear Figure-8 trajectory.

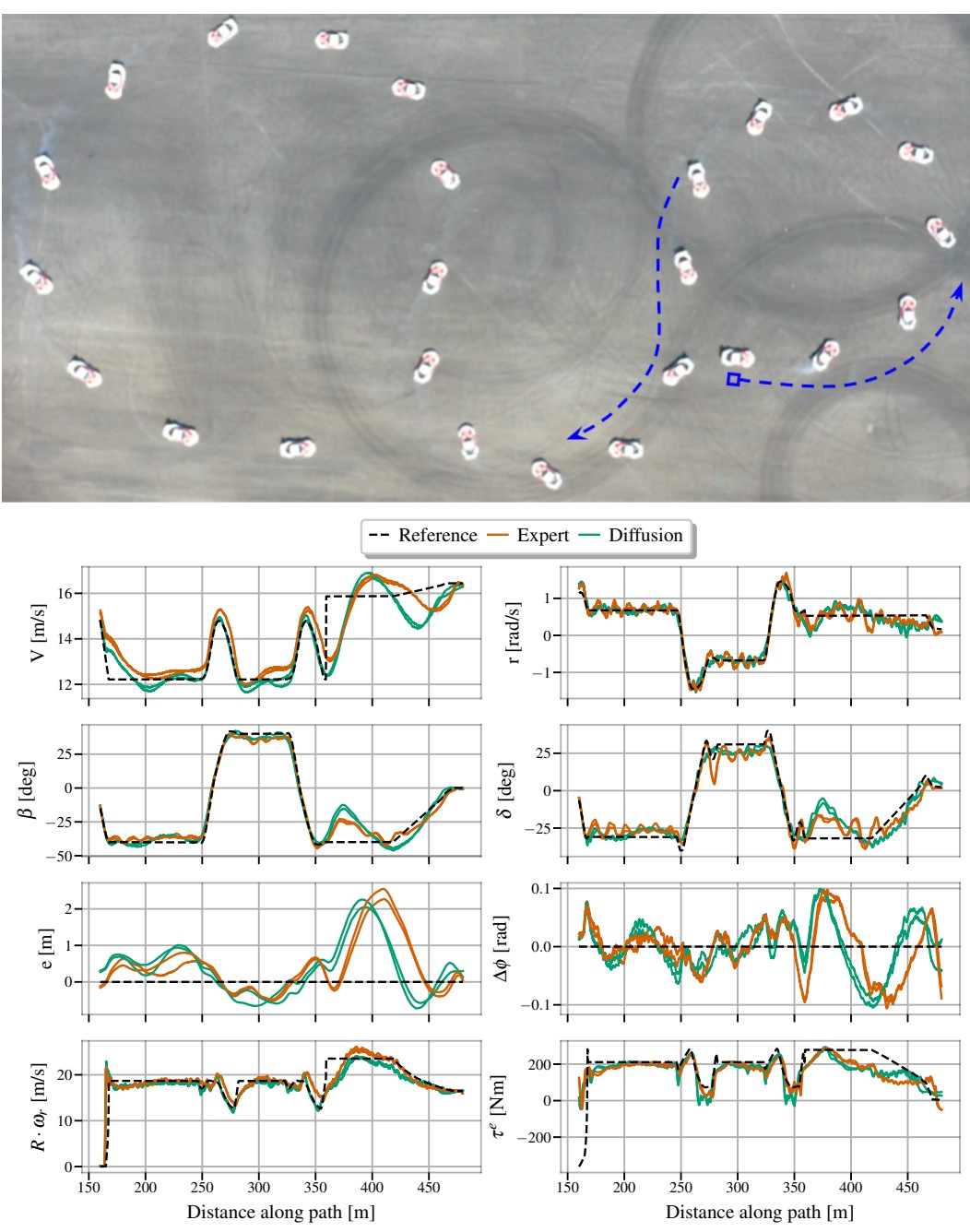

Figure 15: Toyota Supra drifting on a second-gear slalom-like trajectory.

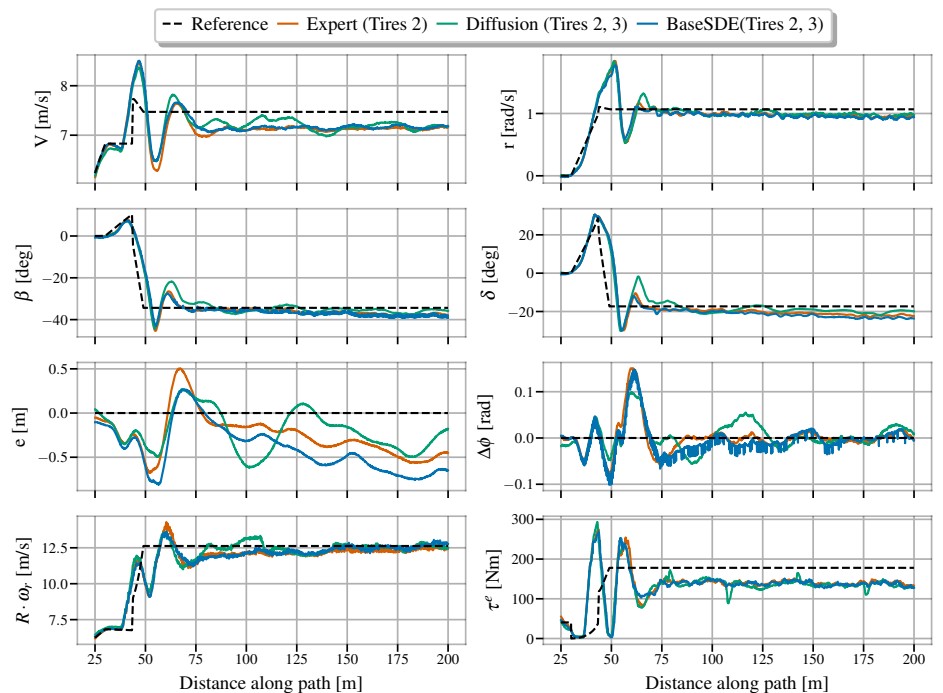

Figure 16: Lexus drifting on a first-gear donut trajectory with Tire 2.

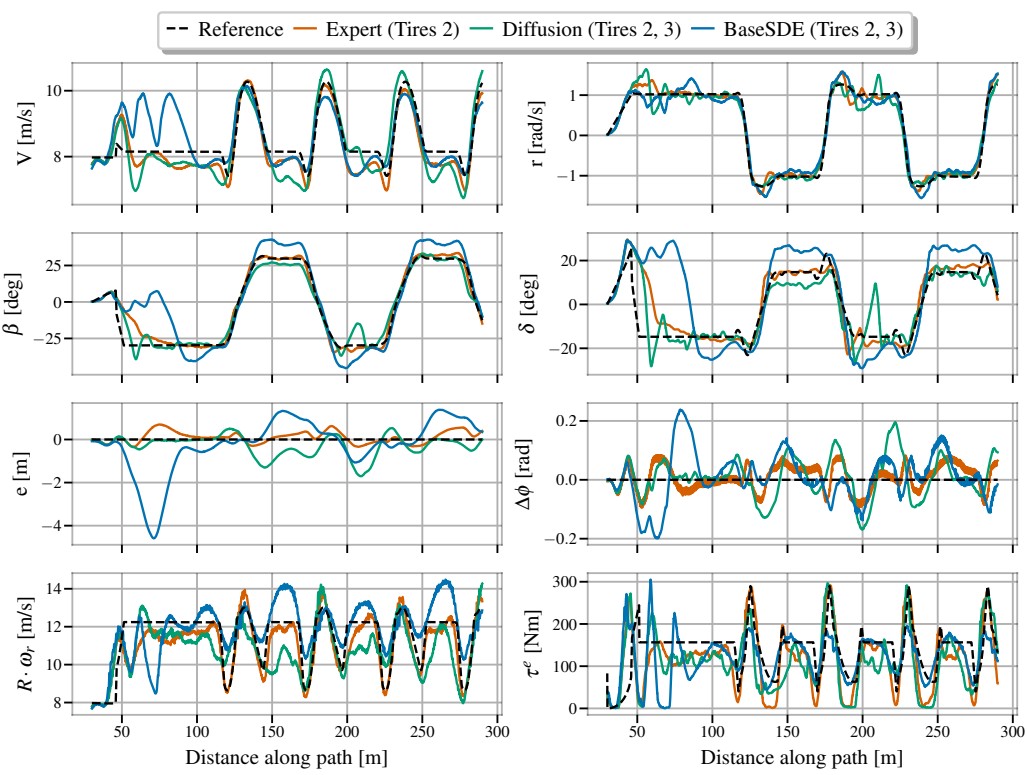

Figure 17: Lexus drifting on a first-gear FIgure-8 trajectory with Tire 2.

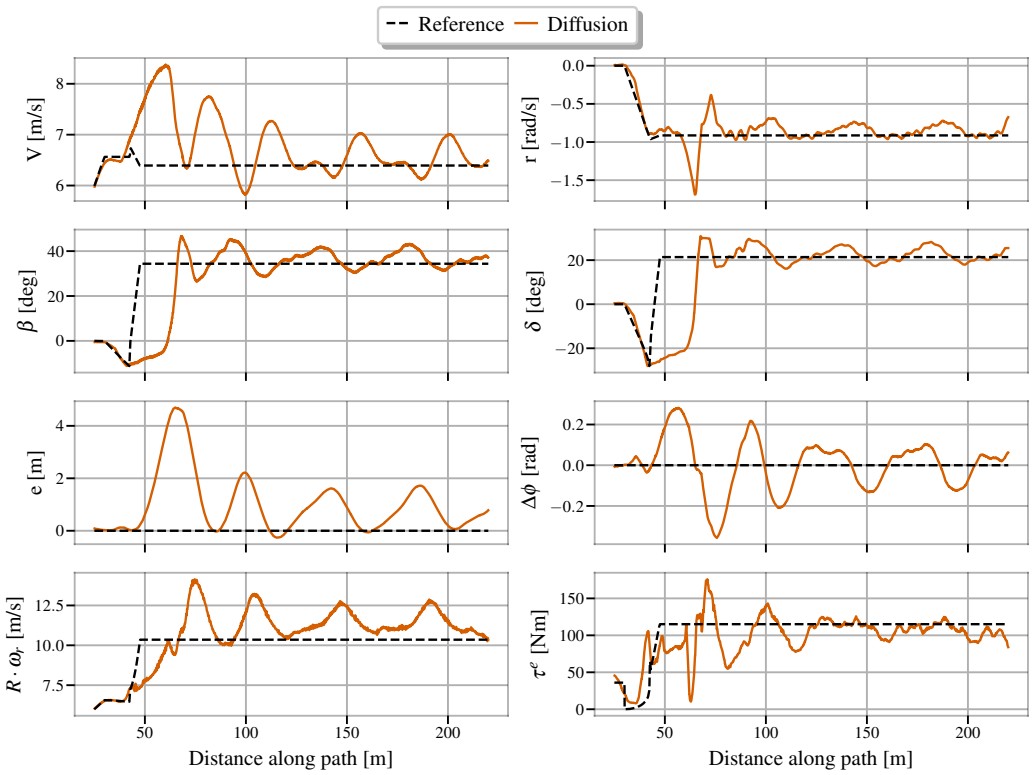

Figure 18: Lexus drifting on a first-gear donut trajectory on heavy rain.

