# OpenReview forum: "One Model to Drift Them All: Physics-Informed Conditional Diffusion Model for Driving at the Limits"
_robot-learning.org/CoRL/2024/Conference — CoRL 2024_

### Official Review · Reviewer_ceJU · 2024-07-20
**One Model to Drift Them All: Physics-Informed Conditional Diffusion Model for Driving at the Limits**

**Originality:** 4
**Technical Quality:** 4
**Clarity Of Presentation:** 3
**Potential Impact:** 3
**Recommendation:** 3
**Confidence:** 3

**Review:**

The work proposed in the paper is quite interesting in that it proposed a method for tackling the task of driving at the limits without access to labeled training data. Experiments demonstrating drifting capabilities were done on 2 different vehicles (Toyota Supra and Lexus LC500).. These vehicles had large differences in their dynamic responses, thereby showing the  extent of their generalization and robustness capabilities.  Experiments were additionally done to test on differing tire, gear and road conditions.

However, at the same time some things are not clear and require further clarification:
The abstract/paper claims that the capability of operating at the limits of driving would improve safety in scenarios like emergency obstacle avoidance. I do not understand how demonstrating drifting/gripping capabilities would imply obstacle avoidance. This was not shown but only claimed in the paper. I reckon this would require additional sensors to first perceive the environment which may then be used to take evasive measure.

The term "multi-modal" has been used in multiple places in the paper in different contexts such as data, uncertainties, trajectories, properties. It is not clear what it precisely means in each context. I only found the explanation for multi-modal trajectory in L101-102.

**Quality Of The Limitations Section:**

3

**Questions For Rebuttal:**

In addition to the questions raised in the previous review section, there are some additional points that require explanation:

1) What are the control inputs (L117) and where do they come from? If they are already available, would this not be considered labeled data?  Also, are the controls not the final output for eventually controlling the vehicle?

2) How fast is the online adaptation of the neural-SDE (L87)?

**Robotics Focus:**

4

**Summary Of Paper:**

The paper proposes a generative model that can handle driving at the limits. A physics informed neural SDE model is introduced for this. Experiments show that the model can perform drifting maneuvers on 2 different vehicles

**Summary Of Recommendation:**

The paper is an interesting read with relevant results for the robotics community. However, there are aspects which are not very clear and require further clarficiation.

---

### Official Review · Reviewer_3p3m · 2024-07-20
**Well written paper on drifting real-world cars with strong and convincing experiments.**

**Originality:** 4
**Technical Quality:** 4
**Clarity Of Presentation:** 3
**Potential Impact:** 3
**Recommendation:** 4
**Confidence:** 4

**Review:**

The paper is well written and addresses a critical point in autonomous driving which is the operation of cars on their limit which is required in extreme situations. However, that is very challenging as we have typically limited data on these situations and the outcome is sensitive to parameters such as friction, ground, tires, etc. The authors overcome this issue in an original way by using an physics informed model where the parameters come from a diffusion model. In this way, the multimodal nature of the problem can be addressed.

The paper presents a strong experimental verification of the proposed method on two real world cars drifting and following desired trajectories.

However, there are some issues that I’d like to see addressed.

-	It’s not clear to me why the limitation of \Sigma in (1) to be a diagonal matrix improves the model’s interpretability. In contrast, understanding the correlations between the states would enhance the understanding.

-	The authors claim the proposed methods can adapt to different tires, drift situations, etc. by observing the past trajectory and then using the diffusion model to give the “best” parameters to the MPC controller as stated in Algorithm 2. However, I don’t understand the relationship between the historic dataset and the T_gen and T_val. Is always the entire dataset T_hist used? If so, how do you deal with the growing size of this set? Would it make sense, to use some kind of sliding window to be able to adapt to suddenly changing situations such as the transition from a paved road to ice?

-	It seems that \Theta^1 in Alg. 2 is the same as \Theta_Best on (9b). If that’s the case, please be consistent with the name.

**Quality Of The Limitations Section:**

3

**Questions For Rebuttal:**

Please address the issues mentioned in the “review” section

**Robotics Focus:**

4

**Summary Of Paper:**

This paper deals with the challenge of operating cars on their limits, i.e. in drifting situations, which is critical for safe autonomous driving in critical situations. The authors propose to use a physics-informed SDE which parameters are learned by a diffusion model. The article shows very strong results in the experimental section where the proposed method is applied to two real-world cars.

**Summary Of Recommendation:**

Paper with strong experimental verification on operating cars in drifting situation which should be accepted at Corl.

---

### Official Review · Reviewer_Xoo7 · 2024-07-21
**A compelling piece of research with intriguing novelty likely underappreciated by the authors themselves**

**Originality:** 4
**Technical Quality:** 3
**Clarity Of Presentation:** 3
**Potential Impact:** 4
**Recommendation:** 4
**Confidence:** 4

**Review:**

The paper is generally well written with numerous places needing correction or improvement (see Questions for Rebuttal). The technical material is complex, the handling is adequate, and the real-world robotics integration is compelling. These in themselves do qualify the paper for acceptance.

However, while the authors framed their work as having to do with a "reliable method" for autonmous driving "at the limits of handling", it seems to this reviewer that (the understanding of) the research could be deepened. Specifically, the work could be understood as about a general architecture for autonomous driving or even robotics and with what has been done so far (about "drifting") as a mere proof of concept -- but a successful one at it. Even more specifically, while the authors called their method "meta learning", it could likely be more appropriately understood in terms of a hierarchical control and decision making architecture. Thus, what's said on Lines 46-47 "the model inference and control loops are decoupled, which enables diffusion sampling at low rates and high-frequency predictive control" could be more easily made sense of. Similarly, the complexity of Algorithm 1 could also likely be understood in terms of co-learning across a hierarchy (concerned with at least two levels). Furthermore, conceptually, we may understand the $\theta$ parameters at the interface between the diffusion model and the neural SDE for MPC as representing "affordances" or "regimes" of affordances, rather than merely as parameter values to the neural SDE. This way, the method is about merging a more semnatically rich representation (from general observation of the environment history, possibly including very complex interaction dynamics -- both tire with ground surface and car with other cars and pedestrians) with a structurally more delineated and thus interpretable control scheme. If under such a general light, the choice of diffusion model and the specific method developed (mostly in Algorithm 1) still make sense, then the contribution of the paper could be truly significant.

**Quality Of The Limitations Section:**

2

**Questions For Rebuttal:**

1. Equation 1: "$dB$" should be "$dW$" to be consistent with Figure 1?

2. Line 125 the variable $z$ seems to have never been introduced.

3. Algorithm 1, Line 6: the condition "if Time to refine" seems to have never been explained.

4. Algorithm 1, Line 10: "Update D with ..." should be something like "Append ... to D" instead? "Update" is too vague.

5. Line 138: "$2W$" -- the variable $W$ was never introduced.

6. Lines 145-146: more explanation?

7. Lines 163-165: this seems quite standard in dealing with diffusion models. Should we just defer to the literature? Otherwise, if there's something new and special, explain!

8. Line 248: "This conclusion ..." -- it's not advicable to do anaphoric reference to the previous section at the beginning of a new section.

9. This reviewer disagrees with the "meta-learning" characterization (Line 86; see Review above for discussion). The authors are encouraged to consider their approach as hierarchical.

10. The authors are also encouraged to discuss Algorithm 1, especially it's complexity from a hierarchical perspective.

11. In terms of limitations, please also consider the method's relevance to handling general complexity of an interactive driving environment (with other road users). Now, in real world handling of driving in snow and ice, it's simply not true that the environment has only the ego car itself. This comment/request is of course also along the recommendation above of considering the method's general relevance.

**Robotics Focus:**

4

**Summary Of Paper:**

The authors designed and trained a diffusion model to map recent trajectory observations into parameter values of a neural SDE (stochastic differential equation) model, which is in turn used in MPC to control a car at the limits of handling (specifically tire dynamics). The trained diffusion model is shown to generalize across vehicle types, weather conditions, and reference trajectory geometry. The authors claim that their main contribution is a "reliable method" for autonmous driving "at the limits of handling".

**Summary Of Recommendation:**

The recommendation is based on the current state of the paper: it introduced a method with novelty to address a meaningful enough challenge (automated car control in static environment at the limits of handling) and demonstrated the effectiveness of the method. The real-world demonstration, especially that of the generalization across car type, weather condition, and trajectory shape, is compelling. The way in which a diffusion model is combined with neural SDE, and through which with MPC, is interesting. That this reviewer thinks the potential of the general architecture goes beyond the authors' own realization certainly adds to the reviewer's own confierence in their recommendation but does not change what's recommended, which again is based on the strength of the current manuscript itself.

---

### Author Rebuttal · Authors · 2024-08-12

We would like to thank the reviewers and area chair for their insightful feedback, which we believe helped to significantly improve the manuscript.

We included the responses to each reviewer’s comments and highlighted the changes in the revised manuscript in blue. Again, we really appreciate the constructive remarks and thank you all for your detailed feedback.

---

### Decision · Program_Chairs · 2024-09-04

**Decision:**

Accept

**Comment:**

The paper presents a method to map recent trajectory observations into parameter values of a neural SDE model via a diffusion model, which is, in turn, used in MPC to control a car at its limits.

Strengths
- The paper presents an interesting approach that enables driving at the limit without access to labeled training data.
- The presented experiments on two different real-world cars are compelling.

Limitations
- The paper needs to provide further clarification on some of the technical details to fully understand its contributions, such as the availability of the control inputs during training, the latency of online adaptation of the neural SDE, etc.
- It would also be helpful to comment on how the method would handle interactive driving scenarios.

**Comments post rebuttal**

The revised paper has satisfactorily addressed the outstanding reviewers' concerns/questions.